# STABLE RANK NORMALIZATION FOR IMPROVED GENERALIZATION IN NEURAL NETWORKS AND GANS

**Amartya Sanyal**
Department of Computer Science
University of Oxford,
The Alan Turing Institute
amartya.sanyal@cs.ox.ac.uk

**Philip H. Torr**
Department of Engineering Science
University of Oxford
philip.torr@eng.ox.ac.uk

**Puneet K. Dokania**
Department of Engineering Science
University of Oxford
puneet@robots.ox.ac.uk

## ABSTRACT

Exciting new work on generalization bounds for neural networks (NN) given by Bartlett et al. (2017); Neyshabur et al. (2018) closely depend on two parameter-dependant quantities a) the Lipschitz constant upper bound and b) the *stable rank* (a softer version of rank). Even though these bounds typically have minimal practical utility, they facilitate questions on whether controlling such quantities together could improve the generalization behaviour of NNs in practice. To this end, we propose *stable rank normalization* (SRN), a novel, provably optimal, and computationally efficient weight-normalization scheme which minimizes the *stable rank* of a linear operator. Surprisingly we find that SRN, despite being *non-convex*, can be shown to have a unique optimal solution. We provide extensive analyses across a wide variety of NNs (DenseNet, WideResNet, ResNet, Alexnet, VGG), where applying SRN to their linear layers leads to improved classification accuracy, while simultaneously showing improvements in generalization, evaluated empirically using shattering experiments (Zhang et al., 2016); and three measures of sample complexity by Bartlett et al. (2017), Neyshabur et al. (2018), & Wei & Ma. Additionally, we show that, when applied to the discriminator of GANs, it improves Inception, FID, and Neural divergence scores, while learning mappings with a low empirical Lipschitz constant.

## 1 INTRODUCTION

Deep neural networks have shown astonishing ability in tackling a wide variety of machine learning problems including a great ability to generalize under extreme over-parameterization. Within this work we leverage very recent, and important, theoretical results on the generalization bounds of deep networks to yield a practical low cost method to normalize the weights within a network using a scheme - which we call Stable Rank Normalization (SRN). The motivation for SRN comes from the generalization bound for NNs given by Neyshabur et al. (2018) and Bartlett et al. (2017),

$\mathcal{O}\sqrt{\prod_i^d \|\mathbf{W}_i\|_2^2 \sum_{i=1}^d \mathrm{srank}(\mathbf{W}_i)}$ [1] , that depend on two parameter-dependent quantities: a) the scale-dependent Lipschitz constant upper-bound $\prod_i^d \|\mathbf{W}_i\|_2$ (product of spectral norms) and b) the sum of scale-independent *stable ranks* ($\mathrm{srank}(\mathbf{W})$). Stable rank is a softer version of the rank operator and is defined as the squared ratio of the Frobenius norm to the spectral norm. Although these two terms appear frequently in these bounds, the empirical impact of simultaneously controlling them on the generalization behaviour of NNs has not been explored yet possibly because of the difficulties associated with optimizing stable rank. This is precisely the goal of this work and based on extensive experiments across a wide variety of NN architectures, we show that, indeed, controlling them

---

[1] $d$ and $\|\mathbf{W}_i\|_2$ represents the number of layers and the spectral norm of the $i$-th linear layer $\mathbf{W}_i$, respectively.

simultaneously improves the generalization behaviour, while improving the classification performance of NNs. We observe improved training of Generative Adversarial Networks (GAN) Goodfellow et al. (2014) as well.

To this end, we propose Stable Rank Normalization (SRN) which allows us to simultaneously control the Lipschitz constant and the stable rank of a linear operator. Note that the widely used Spectral Normalization (SN) (Miyato et al., 2018) allows explicit control over the Lipschitz constant, however, as will be discussed in the paper, it does not have any impact on the stable rank. We would like to emphasize that, as opposed to SN, the SRN solution is optimal and unique even in situations when it is non-convex. It is one of those rare cases where an optimal solution to a provably non-convex problem could be obtained. Computationally, our proposed SRN for NNs is no more complicated than SN, just requiring computation of the largest singular value which can be efficiently obtained using the power iteration method (Mises & Pollaczek-Geiringer, 1929).

**Experiments**   Although SRN is in principle applicable to any problem involving a sequence of affine transformations, considering recent interests, we show its effectiveness when applied to the linear layers of deep neural networks. We perform extensive experiments on a wide variety of NN architectures (DenseNet, WideResNet, ResNet, Alexnet, VGG) for the analyses and show that, SRN, while providing the best classification accuracy (compared against standard, or vanilla, training and SN), consistently shows improvement on the generalization behaviour. We also experiment with GANs and show that, SRN prefers learning discriminators with low empirical Lipschitz while providing improved Inception, FID and Neural Divergence scores (Gulrajani et al., 2019).

We would like to note that although SN is being widely used for the training of GANs, its effect on the generalization behaviour over a wide variety of NNs has not yet been explored. To the best of our knowledge, we are the first to do so.

**Contributions**

- We propose SRN— a novel normalization scheme for simultaneously controlling the Lipschitz constant and the stable rank of a linear operator.
- Optimal and unique solution to the provably non-convex stable rank normalization problem.
- Efficient and easy to implement SRN algorithm for NNs.

## 2   BACKGROUND AND INTUITIONS

**Neural Networks**   Consider $f_\theta : \mathbb{R}^m \to \mathbb{R}^k$ to be a feed-forward multilayer NN parameterized by $\theta \in \mathbb{R}^n$, each layer of which consists of a linear followed by a non-linear[2] mapping. Let $\mathbf{a}_{l-1} \in \mathbb{R}^{n_{l-1}}$ be the input (or pre-activations) to the $l$-th layer, then the output (or activations) of this layer is represented as $\mathbf{a}_l = \phi_l(\mathbf{z}_l)$, where $\mathbf{z}_l = \mathbf{W}_l \mathbf{a}_{l-1} + \mathbf{b}_l$ is the output of the linear (affine) layer parameterized by the weights $\mathbf{W}_l \in \mathbb{R}^{n_{l-1} \times n_l}$ and biases $\mathbf{b}_l \in \mathbb{R}^{n_l}$, and $\phi_l(.)$ is the element-wise non-linear function applied to $\mathbf{z}_l$. For classification tasks, given a dataset with input-output pairs denoted as $(\mathbf{x} \in \mathbb{R}^m, \mathbf{y} \in \{0,1\}^k; \sum_j y_j = 1)$ [3], the parameter vector $\theta$ is learned using back-propagation to optimize the classification loss (*e.g.*, cross-entropy).

**Singular Value Decomposition (SVD)**   Given $\mathbf{W} \in \mathbb{R}^{s \times r}$ with rank $k \le \min(s, r)$, we denote $\{\sigma_i\}_{i=1}^k$, $\{\mathbf{u}_i\}_{i=1}^k$, and $\{\mathbf{v}_i\}_{i=1}^k$ as its singular values, left singular vectors, and right singular vectors, respectively. Throughout this work, a set of singular values is assumed to be sorted $\sigma_1 \ge \cdots \ge \sigma_k$. $\sigma_i(\mathbf{W})$ denotes the $i$-th singular value of the matrix $\mathbf{W}$. Using singular values, the matrix 2-norm $\|\mathbf{W}\|_2$ and the Frobenius norm $\|\mathbf{W}\|_F$ can be computed as $\sigma_1$ and $\sqrt{\sum_i \sigma_i^2}$, respectively.

**Stable Rank**   Below we provide the formal definition and some properties of stable rank.

**Definition 2.1.** *The Stable Rank (Rudelson & Vershynin, 2007) of an arbitrary matrix $\mathbf{W}$ is defined as* $\mathrm{srank}(\mathbf{W}) = \frac{\|\mathbf{W}\|_F^2}{\|\mathbf{W}\|_2^2} = \frac{\sum_{i=1}^k \sigma_i^2(\mathbf{W})}{\sigma_1^2(\mathbf{W})}$, *where $k$ is the rank of the matrix. Stable rank is*

- *a soft version of the rank operator and, unlike rank, is less sensitive to small perturbations.*

---

[2] *e.g.* ReLU, tanh, sigmoid, and maxout.

[3] $y_j$ is the $j$-th element of vector $\mathbf{y}$. Only one class is assigned as the ground-truth label to each $\mathbf{x}$.

- *differentiable as both Frobenius and Spectral norms are almost always differentiable.*
- *upper-bounded by the rank:* $\text{srank}(\mathbf{W}) = \frac{\sum_{i=1}^{k} \sigma_i^2(\mathbf{W})}{\sigma_1^2(\mathbf{W})} \leq \frac{\sum_{i=1}^{k} \sigma_1^2(\mathbf{W})}{\sigma_1^2(\mathbf{W})} = k.$
- *invariant to scaling, implying,* $\text{srank}(\mathbf{W}) = \text{srank}(\frac{\mathbf{W}}{\eta})$, *for any* $\eta \in \mathbb{R} \setminus \{0\}$.

**Lipschitz Constant**  Here we describe the global and the local Lipschitz constants. Briefly, the Lipschitz constant is a quantification of the sensitivity of the output with respect to the change in the input. A function $f : \mathbb{R}^m \mapsto \mathbb{R}^k$ is *globally L-Lipschitz continuous* if $\exists L \in \mathbb{R}_+ : \|f(\mathbf{x}_i) - f(\mathbf{x}_j)\|_q \leq L \|\mathbf{x}_i - \mathbf{x}_j\|_p, \forall (\mathbf{x}_i, \mathbf{x}_j) \in \mathbb{R}^m$, where $\|\cdot\|_p$ and $\|\cdot\|_q$ represents the norms in the input and the output metric spaces, respectively. The global Lipschitz constant $L_g$ is:

$$L_g = \max_{\substack{\mathbf{x}_i, \mathbf{x}_j \in \mathbb{R}^m \\ \mathbf{x}_i \neq \mathbf{x}_j}} \frac{\|f(\mathbf{x}_i) - f(\mathbf{x}_j)\|_q}{\|\mathbf{x}_i - \mathbf{x}_j\|_p}. \tag{1}$$

The above definition of the Lipschitz constant depends on all pairs of inputs in the domain $\mathbb{R}^m \times \mathbb{R}^m$, (thus, global). However, one can define the local Lipschitz constant based on the sensitivity of $f$ in the vicinity of a given point $\mathbf{x}$. Precisely, at $\mathbf{x}$, for an arbitrarily small $\delta > 0$, the local Lipschitz constant is computed on the open ball of radius $\delta$ centered at $\mathbf{x}$. Let $\mathbf{h} \in \mathbb{R}^m$, $\|\mathbf{h}\|_p < \delta$, then, similar to $L_g$, the *local Lipschitz constant* of $f$ at $\mathbf{x}$, $L_l(\mathbf{x})$, is greater than or equal to $\sup_{\mathbf{h} \neq 0, \|\mathbf{h}\|_p < \delta} \frac{\|f(\mathbf{x}+\mathbf{h})-f(\mathbf{x})\|_q}{\|\mathbf{h}\|_p}$. Assuming $f$ to be Fréchet differentiable, as $\mathbf{h} \to 0$, using $f(\mathbf{x} + \mathbf{h}) - f(\mathbf{x}) \approx J_f(\mathbf{x})\mathbf{h}$, $L_l(\mathbf{x})$ is the matrix (operator) norm of the Jacobian $\left( J_f(\mathbf{x}) = \frac{\partial f(\mathbf{z})}{\partial \mathbf{z}}|_{\mathbf{x}} \in \mathbb{R}^{k \times m} \right)$ as follows:[4]

$$L_l(\mathbf{x}) \overset{(a)}{=} \lim_{\delta \to 0} \sup_{\substack{\mathbf{h} \neq 0 \\ \|\mathbf{h}\|_p < \delta}} \frac{\|J_f(\mathbf{x})\mathbf{h}\|_q}{\|\mathbf{h}\|_p} \overset{(b)}{=} \sup_{\substack{\mathbf{h} \neq 0 \\ \mathbf{h} \in \mathbb{R}^m}} \frac{\|J_f(\mathbf{x})\mathbf{h}\|_q}{\|\mathbf{h}\|_p} = \|J_f(\mathbf{x})\|_{p,q}. \tag{2}$$

A function is said to be *locally Lipschitz* with *local Lipschitz constant* $L_l$ if for all $\mathbf{x} \in \mathbb{R}^m$ he function is $L_l$ *locally-Lipschitz* at $\mathbf{x}$. Thus, $L_l = \sup_{\mathbf{x} \in \mathbb{R}^m} L_l(\mathbf{x})$. Notice that the Lipschitz constant (global or local), greatly depends on the chosen norms. When $p = q = 2$, the upperbound on the local Lipschitz constant at $\mathbf{x}$ boils down to the 2-matrix norm (maximum singular value) of the Jacobian $J_f(\mathbf{x})$ (see last equality of (2)). With these preliminary definitions, in Section 3, we discuss more optimistic (or empirical) estimates of $L_l$ and $L_g$, its link with generalization and then in Section 5, we show empirically the effect of SRN on them and on generalization.

**The local Lipschitz upper-bound for Neural Networks**  As mentioned earlier (2), $L_l(\mathbf{x}) = \|J_f(\mathbf{x})\|_{p,q}$, where, in the case of NNs (proof along with why it is loose in Appendix C)

$$L_l(\mathbf{x}) = \|J_f(\mathbf{x})\|_{p,q} \leq \|\mathbf{W}_l\|_{p,q} \cdots \|\mathbf{W}_1\|_{p,q} \quad \text{and} \quad L_l = L_l(\mathbf{x}) \tag{3}$$

## 3  WHY STABLE RANK NORMALIZATION?

**Lipschitz alone is not sufficient**  Although learning low Lipschitz functions has been shown to provide better generalization (Anthony & Bartlett, 2009; Bartlett et al., 2017; Neyshabur et al., 2018; 2015; Yoshida & Miyato, 2017; Gouk et al., 2018), enable stable training of GANs (Arjovsky et al., 2017; Gulrajani et al., 2017; Miyato et al., 2018) and help provide robustness against adversarial attacks (Cisse et al., 2017), controlling Lipschitz upper bound alone is not sufficient to provide assurance on the generalization error. One of the reasons is that it is scale-dependent, implying, for example, even though scaling an entire ReLU network would not alter the classification behaviour, it can massively increase the Lipschitz constant and thus the theoretical generalization bounds. This suggests that either the bound is of no practical utility, or at least one should regulate both—the Lipschitz constant, and the stable rank (scale-independent)—in a hope to see improved generalization in practice.

---

[4]Here, (a) is by definition of local lipschitzness and (b) is due to the property of norms that for any non-negative scalar $c$, $\|c\mathbf{x}\| = c \|\mathbf{x}\|$.

**Stable rank controls the noise-sensitivity**    As shown by Arora et al. (2018), one of the critical properties of generalizable NNs is low noise sensitivity— the ability of a network to preferentially carry over the true signal in the data. For a given noise distribution $\mathcal{N}$, it can be quantified as

$$\Phi_{f_\theta, \mathcal{N}} = \max_{\mathbf{x} \in \mathcal{D}} \Phi_{f_\theta, \mathcal{N}}(\mathbf{x}), \quad where \quad \Phi_{f_\theta, \mathcal{N}}(\mathbf{x}) := \mathbb{E}_{\eta \sim \mathcal{N}} \left[ \frac{\|f_\theta(\mathbf{x} + \eta \|\mathbf{x}\|) - f_\theta(\mathbf{x})\|^2}{\|f_\theta(\mathbf{x})\|^2} \right].$$

For a linear mapping with parameters $\mathbf{W}$ and the noise distribution being normal-$\mathcal{N}(0, \mathbf{I})$, it can be shown that $\Phi_{f_\mathbf{W}, \mathcal{N}} \geq \mathrm{srank}(\mathbf{W})$ (c.f. Proposition 3.1 in Arora et al. (2018)). Thus, decreasing the stable rank directly decreases the lower bound of the noise sensitivity. In Figure 1, we show $\Phi_{f_\theta, \mathcal{N}}$ of a ResNet110 trained on CIFAR100. Note that although the Lipschitz upper bound of SRN and SN are the same, SRN (algorithmic details in Section 4) is much less sensitive to noise as the constraints imposed enforce the stable rank to decrease to 30% of its original value, which in effect reduces the noise sensitivity.

**Stable rank impacts empirical Lipschitz constant**    It is apparent that the Lipschitz constant upper bound ($\prod_i^d \|\mathbf{W}_i\|_2$), along with being scale-dependent, also is *data-independant* and hence, provides a pessimistic estimate of the behaviour of a model on a particular task or dataset. Considering this, a relatively optimistic estimate of the model's behaviour would be an *empirical* estimate of the Lipschitz constant ($L_e$) on a task-specific dataset [5]. Note that local $L_e$ is just the norm of the Jacobian at a given point [6]. Empirically, Novak et al. (2018) provided results showing how local $L_e$ (in the vicinity of train data) is correlated with the generalization error of NNs. This observation is further supported by the work of Wei & Ma; Nagarajan & Kolter (2019); Arora et al. (2018) whereby the variants of $L_e$ are used to derive generalization bounds. Thus, a tool that favours low $L_e$ is likely to provide better generalization behaviour in practice. To this end, we first consider a simple two layer linear-NN example in Appendix B.2 and show that low rank mappings do favour low

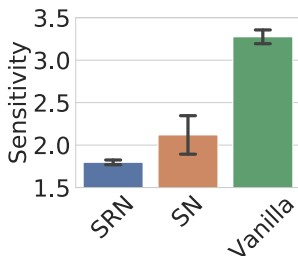

Figure 1: Noise Sensitivity (lower the better). Test accuracy: SRN (73.1%), SN (71.5%), and Vanilla (72.4%).

$L_e$. Since direct minimization of rank for NNs is non-trivial, the expectation is that learning low stable rank (softer version of rank) might induce similar behaviour. We experimentally validate this hypothesis by showing that, as we decrease the stable rank, the empirical Lipschitz decreases. This shows SRN indeed prefers mappings with a low empirical Lipschitz constant.

# 4    STABLE RANK NORMALIZATION

Here we provide a theoretically sound procedure to do SRN. A big challenge in stable rank normalization comes from the fact that it is scale-invariant (refer Definition 2.1), thus, any normalization scheme that modifies $\mathbf{W} = \sum_i \sigma_i \mathbf{u}_i \mathbf{v}_i^\top$ to $\widehat{\mathbf{W}} = \sum_i \frac{\sigma_i}{\eta} \mathbf{u}_i \mathbf{v}_i^\top$ will have no effect on the stable rank, making SRN non-trivial. Examples of such schemes are SN (Miyato et al., 2018) where $\eta = \sigma_1$, and Frobenius normalization where $\eta = \|\mathbf{W}\|_\mathrm{F}$. As will be shown, our approach to stable rank normalization is optimal and efficient. Note, the widely used SN (Miyato et al., 2018) is not optimal (proof in Appendix A.2).

**The SRN Problem Statement**    Given a matrix $\mathbf{W} \in \mathbb{R}^{m \times n}$ with rank $p$ and spectral partitioning index $k$ ($0 \leq k < p$), we formulate the SRN problem as:

$$\underset{\widehat{\mathbf{W}}_k \in \mathbb{R}^{m \times n}}{\arg \min} \left\| \mathbf{W} - \widehat{\mathbf{W}}_k \right\|_\mathrm{F}^2 \quad s.t. \quad \underbrace{\mathrm{srank}(\widehat{\mathbf{W}}_k) = r}_{\text{stable rank constraint}}, \underbrace{\lambda_i = \sigma_i, \forall i \in \{1, \cdots, k\}}_{\text{spectrum preservation constraints}}. \tag{4}$$

where, $1 \leq r < \mathrm{srank}(\mathbf{W})$ is the desired stable rank, $\lambda_i$s and $\sigma_i$s are the singular values of $\widehat{\mathbf{W}}_k$ and $\mathbf{W}$, respectively. The partitioning index $k$ is used for the *singular value (or the spectrum) preservation*

---

[5]It can be the train-/test- data, the generated data (*e.g.*, in GANs), or some interpolated data points.

[6]For completeness, we provide the relationship between the global and the local $L_e$ in Proposition B.1.

*constraint*. It gives us the flexibility to obtain $\widehat{\mathbf{W}}_k$ such that its top $k$ singular values are exactly the same as that of the original matrix. Note, the problem statement is more general in the sense that putting $k = 0$ removes the spectrum preservation constraint.

**The Solution to SRN** The optimal unique solution to the above problem is provided in Theorem 1 and proved in Appendix A.1. Note, at $k = 0$, the problem (4) is non-convex, otherwise convex.

**Theorem 1.** *Given a real matrix* $\mathbf{W} \in \mathbb{R}^{m \times n}$ *with rank* $p$, *a target spectrum (or singular value) preservation index* $k$ *(*$0 \leq k < p$*), and a target stable rank of* $r$ *(*$1 \leq r < \text{srank}(\mathbf{W})$*), the optimal solution* $\widehat{\mathbf{W}}_k$ *to problem (4) is* $\widehat{\mathbf{W}}_k = \gamma_1 \mathbf{S}_1 + \gamma_2 \mathbf{S}_2$, *where* $\mathbf{S}_1 = \sum_{i=1}^{\max(1,k)} \sigma_i \mathbf{u}_i \mathbf{v}_i^\top$ *and* $\mathbf{S}_2 = \mathbf{W} - \mathbf{S}_1$. $\{\sigma_i\}_{i=1}^k$, $\{\mathbf{u}_i\}_{i=1}^k$ *and* $\{\mathbf{v}_i\}_{i=1}^k$ *are the top* $k$ *singular values and vectors of* $\mathbf{W}$, *and, depending on* $k$, $\gamma_1$ *and* $\gamma_2$ *are defined below. For simplicity, we first define* $\gamma = \frac{\sqrt{r\sigma_1^2 - \|\mathbf{S}_1\|_F^2}}{\|\mathbf{S}_2\|_F}$, *then*

*a) If* $k = 0$ *(no spectrum preservation), the problem becomes non-convex, the optimal solution to which is obtained for* $\gamma_2 = \frac{\gamma + r - 1}{r}$ *and* $\gamma_1 = \frac{\gamma_2}{\gamma}$, *when* $r > 1$. *If* $r = 1$, *then* $\gamma_2 = 0$ *and* $\gamma_1 = 1$. *Since, in this case,* $\|\mathbf{S}_1\|_F^2 = \sigma_1^2$, $\gamma = \frac{\sqrt{r-1}\sigma_1}{\|\mathbf{S}_2\|_F}$.

*b) If* $k \geq 1$, *the problem is convex. If* $r \geq \frac{\|\mathbf{S}_1\|_F^2}{\sigma_1^2}$ *the optimal solution is obtained for* $\gamma_1 = 1$, *and* $\gamma_2 = \gamma$ *and if not, the problem is not feasible.*

*c) Also,* $\left\|\widehat{\mathbf{W}}_k - \mathbf{W}\right\|_F$ *monotonically increases with* $k$ *for* $k \geq 1$.

Intuitively, Theorem 1 partitions the given matrix into two parts, depending on $k$, and then scales them differently in order to obtain the optimal solution. The value of the partitioning index $k$ is a design choice. If there is no particular preference to $k$, then $k = 0$ provides the most optimal solution. We provide a simple example to better understand this. Given $\mathbf{W} = \mathbb{I}_3$ (rank = srank($\mathbf{W}$) = 3), the objective is to project it to a new matrix with stable rank of 2. Solutions to this problem are:

$$\widehat{\mathbf{W}}_1 = \begin{bmatrix} 1 & 0 & 0 \\ 0 & 1 & 0 \\ 0 & 0 & 0 \end{bmatrix}, \ \widehat{\mathbf{W}}_2 = \begin{bmatrix} 1 & 0 & 0 \\ 0 & \frac{1}{\sqrt{2}} & 0 \\ 0 & 0 & \frac{1}{\sqrt{2}} \end{bmatrix}, \ \widehat{\mathbf{W}}_3 = \begin{bmatrix} \frac{\sqrt{2}+1}{2} & 0 & 0 \\ 0 & \frac{\sqrt{2}+1}{2\sqrt{2}} & 0 \\ 0 & 0 & \frac{\sqrt{2}+1}{2\sqrt{2}} \end{bmatrix} \tag{5}$$

Here, $\widehat{\mathbf{W}}_1$ is obtained using the standard rank minimization (Eckart-Young-Mirsky (Eckart & Young, 1936)) while $\widehat{\mathbf{W}}_2$ and $\widehat{\mathbf{W}}_2$ are the solutions of Theorem 1 with $k = 1$ and $k = 0$, respectively. It is easy to verify that the stable rank of all the above solutions is 2. However, the Frobenius distance (lower the better) of these solutions from the original matrix follows the order $\left\|\mathbf{W} - \widehat{\mathbf{W}}_1\right\|_F > \left\|\mathbf{W} - \widehat{\mathbf{W}}_2\right\|_F > \left\|\mathbf{W} - \widehat{\mathbf{W}}_3\right\|_F$. As evident from the example, the solution to SRN, instead of completely removing a particular singular value, scales them (depending on $k$) such that the new matrix has the desired stable rank. Note that for $\widehat{\mathbf{W}}_1$ (true for any $k \geq 1$), the spectral norm of the original and the normalized matrices are the same, implying, $\gamma_1 = 1$. However, for $k = 0$, the spectral norm of the optimal solution is greater than that of the original matrix. It is easy to verify from Theorem 1 that as $k$ increases, $\gamma_2$ decreases. Thus, the amount of scaling required for the second partition $\mathbf{S}_2$ is more aggressive. In all situations, the following inequality holds: $\gamma_2 \leq 1 \leq \gamma_1$.

---

**Algorithm 1** Stable Rank Normalization

**Require:** $\mathbf{W} \in \mathbb{R}^{m \times n}, r, k \geq 1$
1: $\mathbf{S}_1 \leftarrow \mathbf{0}, \beta \leftarrow \|\mathbf{W}\|_F^2, \eta \leftarrow 0, l \leftarrow 0$
2: **for** $i \in \{1, \cdots, k\}$ **do**
3: $\quad \{\mathbf{u}_i, \mathbf{v}_i, \sigma_i\} \leftarrow SVD(\mathbf{W}, i)$
4: $\quad \triangleright$ Power method to get $i$-th singular value
5: $\quad$ **if** $r \geq (\sigma_i^2 + \eta)/\sigma_1^2$ **then**
6: $\quad\quad \mathbf{S}_1 \leftarrow \mathbf{S}_1 + \sigma_i \mathbf{u}_i \mathbf{v}_i^\top$
7: $\quad\quad \eta \leftarrow \eta + \sigma_i^2, \beta \leftarrow \beta - \sigma_i^2$
8: $\quad\quad l \leftarrow l + 1$
9: $\quad$ **else**
10: $\quad\quad$ break
11: $\quad$ **end if**
12: **end for**
13: $\eta \leftarrow r\sigma_1^2 - \eta$
14: **return** $\widehat{\mathbf{W}}_l \leftarrow \mathbf{S}_1 + \sqrt{\frac{\eta}{\beta}}(\mathbf{W} - \mathbf{S}_1), l$

**Algorithm 2** SRN for a Linear Layer in NN

**Require:** $\mathbf{W} \in \mathbb{R}^{m \times n}, r,$ learning rate $\alpha$, mini-batch dataset $\mathcal{D}$
1: Initialize $\mathbf{u} \in \mathbb{R}^m$ with a random vector.
2: $\mathbf{v} \leftarrow \frac{\mathbf{W}^\top \mathbf{u}}{\|\mathbf{W}^\top \mathbf{u}\|}, \mathbf{u} \leftarrow \frac{\mathbf{W}^\top \mathbf{v}}{\|\mathbf{W}^\top \mathbf{v}\|}$
3: $\quad\quad\quad \triangleright$ Perform power iteration
4: $\sigma(\mathbf{W}) = \mathbf{u}^\top \mathbf{W} \mathbf{v}$
5: $\mathbf{W}_f = \mathbf{W}/\sigma(\mathbf{W}) \triangleright$ Spectral Normalization
6: $\widehat{\mathbf{W}} = \mathbf{W}_f - \mathbf{u}\mathbf{v}^\top$
7: **if** $\left\|\widehat{\mathbf{W}}\right\|_F \leq \sqrt{r-1}$ **then**
8: $\quad$ **return** $\mathbf{W}_f$
9: **end if**
10: $\mathbf{W}_f = \mathbf{u}\mathbf{v}^\top + \widehat{\mathbf{W}} \frac{\sqrt{r-1}}{\|\widehat{\mathbf{w}}\|_F} \quad\quad \triangleright$ Stable Rank Normalization
11: **return** $\mathbf{W} \leftarrow \mathbf{W} - \alpha \nabla_\mathbf{W} L(\mathbf{W}_f, \mathcal{D})$

---

**Algorithm for Stable Rank Normalization** We provide a general procedure in Algorithm 1 to solve the stable rank normalization problem for $k \geq 1$ (the solution for $k = 0$ is straightforward from Theorem 1). Claim 2 provides the properties of the algorithm. The algorithm is constructed so that the prior knowledge of the rank of the matrix is not necessary.

**Claim 2.** *Given a matrix* $\mathbf{W}$*, the desired stable rank* $r$*, and the partitioning index* $k \geq 1$*, Algorithm 1 requires computing the top* $l$ *($l \leq k$) singular values and vectors of* $\mathbf{W}$*. It returns* $\widehat{\mathbf{W}}_l$ *and the scalar* $l$ *such that* $\mathrm{srank}(\widehat{\mathbf{W}}_l) = r$*, and the top* $l$ *singular values of* $\mathbf{W}$ *and* $\widehat{\mathbf{W}}_l$ *are the same. If* $l = k$*, then the solution provided is the optimal solution to the problem* (4) *with all the constraints satisfied, otherwise, it returns the largest* $l$ *up to which the spectrum is preserved.*

**Combining Stable Rank and Spectral Normalization for NNs** Following the arguments provided in Section 1 and 3, for better generalizability, we propose to normalize *both* the stable rank and the spectral norm of each linear layer of a NN simultaneously. To do so, we first perform approximate SN (Miyato et al., 2018), and then perform optimal SRN (using Algorithm 1). We use $k = 1$ to ensure that the first singular value (which is now normalized) is preserved. Algorithm 2 provides a simplified procedure for the same for a given linear layer of a NN. Note, the computational cost of this algorithm is *exactly the same as that of SN*, which is to compute the top singular value using the power iteration method.

## 5 EXPERIMENTS

**Dataset and Architectures** For classification, we perform experiments on ResNet-110 (He et al., 2016), WideResNet-28-10 (Zagoruyko & Komodakis, 2016), DenseNet-100 (Huang et al., 2017), VGG-19 (Simonyan & Zisserman, 2014), and AlexNet (Krizhevsky & Hinton, 2009) using the CIFAR100 (Krizhevsky & Hinton, 2009) dataset. We present further experiments with CIFAR10 in Appendix D.1 in Figure 10 and 11. We train them using standard training recipes with SGD, using a learning rate of 0.1 (except AlexNet where we use a learning rate of 0.01), and a momentum of 0.9 with a batch size of 128 (further details in Appendix D). In addition to training for a fixed number of epochs, we also present results in the Appendix in Figure 8 and 9 where the training accuracy (as opposed to number of iterations) is used as a stopping criterion to show that our regularizor performs well with a range of stopping criterions.

For GAN experiments, we use CIFAR100, CIFAR10, and CelebA (Liu et al., 2015) datasets. We show results on both, conditional and unconditional GANs. Please refer to Appendix E.1 for further details about the training setup.

**Choosing stable rank** Given a matrix $\mathbf{W} \in \mathbb{R}^{m \times n}$, the desired stable rank $r$ is controlled using a single hyperparameter $c$ as $r = c \min(m, n)$, where $c \in (0, 1]$. For simplicity, we use the same $c$ for

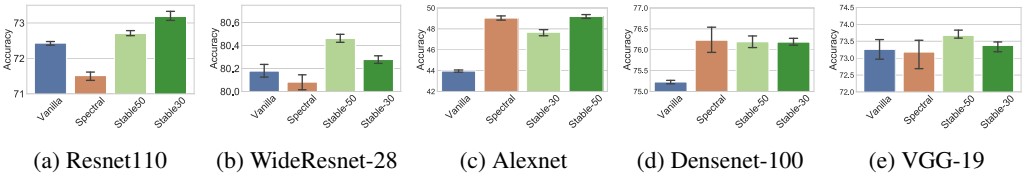

(a) Resnet110        (b) WideResnet-28        (c) Alexnet        (d) Densenet-100        (e) VGG-19

Figure 2: Test accuracies on CIFAR100 for clean data. Higher is better.

all the linear layers. It is trivial to note that if $c = 1$, or for a given $c$, if $\mathrm{srank}(\mathbf{W}) \leq r$, then SRN boils down to SN. For classification, we choose $c = \{0.3, 0.5\}$, and compare SRN against standard training (Vanilla) and training with SN. For GAN experiments, we choose $c = \{0.1, 0.3, 0.5, 0.7, 0.9\}$, and compare SRN-GAN against SN-GAN (Miyato et al., 2018), WGAN-GP (Gulrajani et al., 2017), and orthonormal regularization GAN (Ortho-GAN) (Brock et al., 2016).

**Result Overview**

- SRN improves classification accuracy on a wide variety of architectures.
- Normalizing stable rank improves the learning capacity of spectrally normalized networks.
- SRN shows remarkably less memorization, even on settings very hard to generalize.
- SRN shows much improved generalization behaviour evaluated using recently proposed sample complexity measures.
- As we decrease the stable rank, the empirical Lipschitz of SRN-GAN decreases. Proving our arguments provided in Section 3.
- SRN-GAN provides much improved Neural divergence score (ND) (Gulrajani et al., 2019) compared to SN-GAN, proving that it is robust to memorization in GANs as well.
- SRN-GAN also provide improved Inception and FID scores in all our experiments (except one where SN-GAN is better).

### 5.1   CLASSIFICATION EXPERIMENTS

We perform each experiment $5$ times using a new random seed each time and report the mean, and the $75\%$ confidence interval for the test error in Figure 2. These experiments show that the test accuracy of SRN, on a wide variety NNs, is always higher than the Vanilla and SN (except for SRN-50 on Alexnet where SRN and SN are almost equal). However, SN performs slightly worse than Vanilla for WideResNet-28 and ResNet110. The fact that SRN does involve SN, combined with the above statement, indicate that even though SN reduced the learning capability of these networks, normalizing stable rank must have improved it significantly in order for SRN to outperform Vanilla. For example, in the case of ResNet110, SN is $71.5\%$ accurate whereas SRN provides an accuracy of $73.2\%$. In addition to this, we would like to note that even though SN is being used extensively for the training of GANs, it is not a popular choice when it comes to training standard NNs for classification. We suspect that this is because of the decrease in the capacity, which we have shown to be increased by the stable rank normalization, proving the worth of SRN for classification tasks as well.

### 5.2   STUDY OF GENERALIZATION BEHAVIOUR

Our last set of experiments established that SRN provides improved classification accuracies on various NNs. Here we study the generalization behaviour of these models. Quantifying generalization behaviour is non-trivial and there is no clear answer to it. However, we utilize recent efforts that explore the theoretical understanding of generalization and use them to study it in practice.

**Shattering Experiments**   To inspect the generalization behaviour in NNs we begin with the shattering experiment (Zhang et al., 2016). It is a test of whether the network can fit the training data well but not a label-randomized version of it (each image of the dataset is assigned a random label). As there is no correlation of the labels with the data points $P(y|\mathbf{x})$ is essentially uninformative because it is uniformly random. Thus, the test accuracy on this task is almost $1\%$. A high training accuracy — which indicates a high generalization gap (difference between train and test accuracy) can be achieved

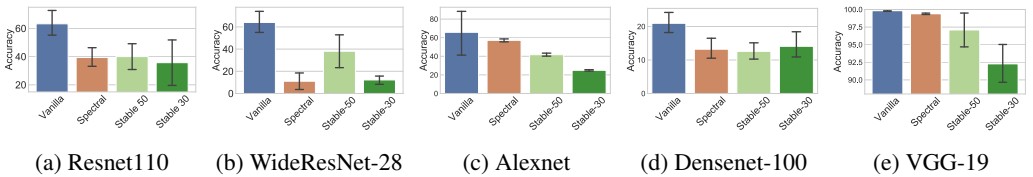

(a) Resnet110     (b) WideResNet-28     (c) Alexnet     (d) Densenet-100     (e) VGG-19

Figure 3: Train accuracies on CIFAR100 for shattering experiment. Lower indicate less memorization, thus, better.

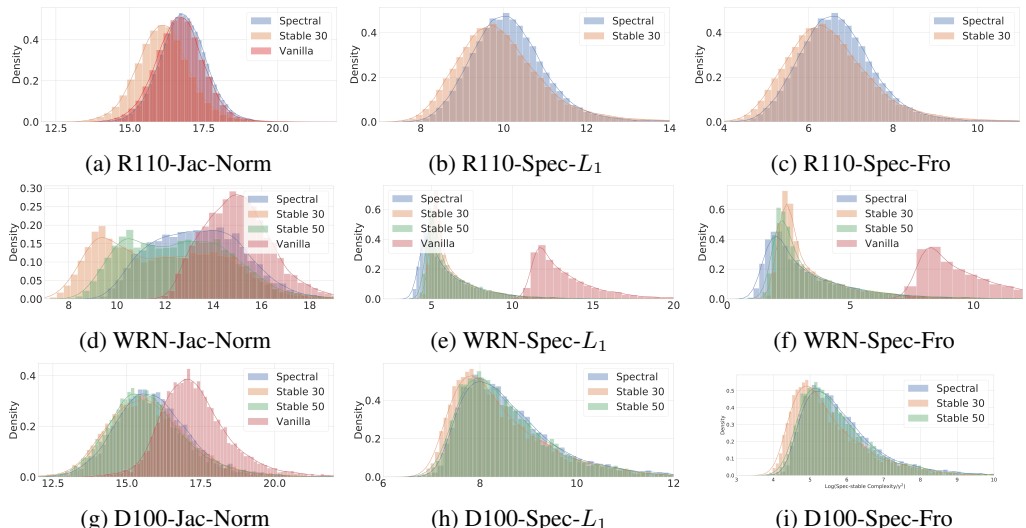

(a) R110-Jac-Norm     (b) R110-Spec-$L_1$     (c) R110-Spec-Fro

(d) WRN-Jac-Norm     (e) WRN-Spec-$L_1$     (f) WRN-Spec-Fro

(g) D100-Jac-Norm     (h) D100-Spec-$L_1$     (i) D100-Spec-Fro

Figure 4: (log) Sample complexity ($C_{\mathrm{alg}}$) of ResNet-110 (Figure 4a to 4c), WideResNet-28-10 (Figure 4d to 4f), and Densenet-100 (Figure 4g to 4i) quantified using the three measures discussed in the paper. Left is better. Vanilla is omitted from Figure 4b, 4c, 4h and 4i as it is too far to the right. Also, in situations where SRN-50 and SN performed the same, we removed the histogram to avoid clutter.

only by memorizing the train data [7]. Figure 3 shows that SRN reduces memorization on random labels (thus, reduces the estimate of the Rademacher complexity (Zhang et al., 2016)). Note, as shown in the classification experiments, the same model was able to achieve the highest training accuracy when the labels were not randomized.

| | SRN-50 | SRN-30 | Spectral (SN) | Vanilla |
|---|---|---|---|---|
| WD | $12.02 \pm 1.77$ | $11.87 \pm 0.57$ | $11.13 \pm 2.56$ | $10.56 \pm 2.32$ |
| w/o WD | $17.71 \pm 2.30$ | $19.04 \pm 4.53$ | $17.22 \pm 1.94$ | $13.49 \pm 1.93$ |

Table 1: **Highly non-generalizable setting**. Training error for ResNet-110 on CIFAR100 with randomized labels, low lr= 0.01, and with and without weight decay. (Higher is better.) The clean test accuracy for this setting is shown in Appendix D.1.

We also look specifically at highly non-generalizable settings — *low learning rate and without weight decay*. As shown in Table 1, SRN consistently achieves lower generalization error (by achieving a low train error) both in the presence and the absence of weight decay [8]. Similar results are reported for Alexnet and WideResNet in Appendix D.1.

---

[7]The training of all the models of one architecture were stopped after the same number of epochs - double the number of epochs the model were trained on the clean dataset.

[8]These result are reported after 200 epochs. It can be looked on as combined with early stopping, which is a powerful way of avoiding memorizing random labels (Li et al.).

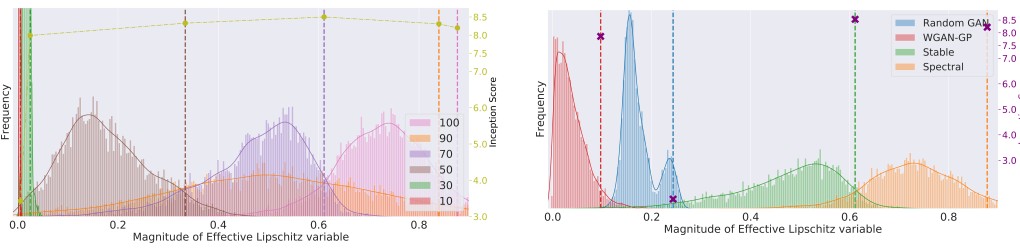

(a) Varying stable rank constraints    (b) Comparison against different approaches

Figure 5: **eLhist** for unconditional GAN on CIFAR10. Dashed vertical lines represent 95*th* percentile. Solid circles and crosses represent the *inception score* for each histogram. Figure 5a shows SRN-GAN for different stable rank constraints (*e.g.* 90 implies $c = 0.9$). Figure 5b compares various approaches. Random-GAN represents random initialization (no training). For SRN-GAN, we use $c = 0.7$.

**Empirical Evaluation of Generalization Behaviour**    When all the factors in training (eg. architecture, dataset, optimizer, among other) as in SRN vs SN vs Vanilla, are fixed, and the only variability is in the normalization, the generalization error can be written as $|\text{Train Err} - \text{Test Err}| \leq \widetilde{\mathcal{O}}\left(\sqrt{C_{\text{alg}}/m}\right)$ where $\widetilde{\mathcal{O}}(\cdot)$ ignores the logarithmic terms, $m$ is the number of samples in the dataset, and $C_{\text{alg}}$ denotes a measure of *sample complexity* for a given algorithm. Lower the value of $C_{\text{alg}}$, the better is the generalization. Before we give various expressions for $C_{\text{alg}}$, we first define a common quantity in all these expressions, called the *margin* $\gamma = f_\theta(\mathbf{x})[y] - \max_{j \neq y} f_\theta(\mathbf{x})[j]$. It measures the gap between the output of the network on the correct label and the other labels. Now we define three recently proposed sample complexity measures useful to quantify the generalization behaviour with further descriptions in Appendix D.1:

- **Spec-Fro:** $\prod_{i=1}^{L} \|\mathbf{W}_i\|_2^2 \sum_{i=1}^{L} \text{srank}(\mathbf{W}_i)/\gamma^2$ (Neyshabur et al., 2018).

- **Spec-L1:** $\prod_{i=1}^{L} \|\mathbf{W}_i\|_2^2 \left(\sum_{i=1}^{L} \frac{\|\mathbf{w}_i\|_{2,1}^{2/3}}{\|\mathbf{w}_i\|_2^{2/3}}\right)^3/\gamma^2$ (Bartlett et al., 2017), $\|.\|_{2,1}$ is the matrix 2-1 norm.

- **Jac-Norm:** $\sum_{i=1}^{L} \|\mathbf{h}_i\|_2 \|\mathbf{J}_i\|_2/\gamma$ (Wei & Ma), where $\mathbf{h}_i$ is the $i^{th}$ hidden layer and $\mathbf{J}_i = \frac{\partial \gamma}{\partial h_i}$

**Histogram of the Empirical Lipschitz Constant (eLhist)**    We evaluate above mentioned sample complexity measures on $10,000$ points from the dataset and plot the distribution of the $\log$ using a histogram shown in Figure 4. The more to the left the histogram, the better is the generalization capacity of the network.

For better clarity, we provide the 90 percentile for each of these histograms in Table 5 in Appendix D.1. As the plots and the table show, both SRN and SN produces a much smaller quantity than a Vanilla network and in 7 out of the 9 cases, SRN is better than SN. The difference between SRN and SN is much more significant in the case of Jac-Norm. As this depend on the empirical lipschitzness, it provides the empirical validation of our arguments in Section 3.

*Above experiments indicate that SRN, while providing enough capacity for the standard classification task, is remarkably less prone to memorization and provides improved generalization.*

5.3    TRAINING OF GENERATIVE ADVERSARIAL NETWORKS (SRN-GAN)

In GANs, there is a natural tension between the *capacity* and the *generalizability* of the discriminator. The capacity ensures that if the generated distribution and the data distribution are different, the discriminator has the ability to distinguish them. At the same time, the discriminator has to be generalizable, implying, the class of hypothesis should be small enough to ensure that it is not just memorizing the dataset. Based on these arguments, we use SRN in the discriminator of GAN which we call SRN-GAN, and compare it against SN-GAN, WGAN-GP, and orthonormal regularization based GAN (Ortho-GAN).

Along with providing results using evaluation metrics such as Inception score (IS) (Salimans et al., 2016) , FID (Heusel et al., 2017), and Neural divergence score (ND) (Gulrajani et al., 2019), we use histograms of the empirical Lipschitz constant, *refered to as eLhist* from now onwards,

---

[1]Results are taken from Miyato et al. (2018). The rest of the results in the tables are generated by us.

for the purpose of analyses. For a given trained GAN (unconditional), we create $2,000$ pairs of samples, where each pair $(\mathbf{x}_i, \mathbf{x}_j)$ consists of $\mathbf{x}_i$ (randomly sampled from the 'real' dataset) and $\mathbf{x}_j$ (randomly sampled from the generator). Each pair is then passed through the discriminator to compute $\|f(\mathbf{x}_i) - f(\mathbf{x}_j)\|_2 / \|\mathbf{x}_i - \mathbf{x}_j\|_2$, which we then use to create the histogram. In the conditional setting, we sample a class from a discrete uniform distribution, and then follow the same approach as described for the unconditional setting.

**Effect of Stable Rank on eLhist and Inception Score**  As shown in Figure 5a, lowering the value of $c$ (aggressive reduction in the stable rank) moves the histogram towards zero, implying, lower empirical Lipschitz constant. This validates our arguments provided in Section 3. Lowering $c$ also improves inception score, however, extreme reduction in the stable rank ($c = 0.1$) dramatically collapses the histogram to zero and also drops the inception score significantly. This is due to the fact that at $c = 0.1$, the capacity of the discriminator is reduced to the point that it is not able to learn to differentiate between the real and the fake samples anymore.

|  | Algorithm | Inception Score | FID | Intra-FID |
|---|---|---|---|---|
| Uncond. | Orthonormal[1] | $7.92 \pm .04$ | 23.8 | - |
| | WGAN-GP | $7.86 \pm .07$ | 21.7 | - |
| | SN-GAN[1] | $8.22 \pm .04$ | 20.67 | - |
| | SRN-70-GAN | $\mathbf{8.53} \pm 0.04$ | **19.83** | - |
| | SRN-50-GAN | $8.33 \pm 0.06$ | **19.57** | - |
| Cond. | SN-GAN | $8.71 \pm .04$ | $16.04^9$ | 26.24 |
| | SRN-70-GAN | $\mathbf{8.93} \pm 0.12$ | **15.92** | **24.01** |
| | SRN-50-GAN | $8.76 \pm 0.09$ | 16.89 | 27.3 |

Table 2: Inception and FID score on CIFAR10.

Table 2 and 3 show that SRN-GAN consistently provide better FID score and an extremely competitive inception score on CIFAR10 (both conditional and unconditional setting) and CIFAR100 (unconditional setting). In Table 4, we compare the ND loss on CIFAR10 and CelebA datasets. Note, ND has been looked as a metric *more robust to memorization* than FID and IS in recent works (Gulrajani et al., 2019; Arora & Zhang, 2017). We report our exact setting to compute ND in Appendix E.1. We essentially report the loss incurred by a *fresh* classifier trained to discriminate the generator distribution and the data distribution. Thus higher the loss, the better the generated images. As evident, SRN-GAN provides better ND scores on both datasets. For a qualitative analysis of the images, we compare generations in both conditional and unconditional setting in Appendix F.

**Comparing different approaches**  In addition, in Figure 5b, we provide eLhist for comparing different approaches. Random-GAN, as expected, has a low empirical Lipschitz constant and extremely poor inception score. Unsurprisingly, WGAN-GP has a lower $L_e$ than Random-GAN, due to its explicit constraint on the Lipschitz constant, while providing a higher inception score. On the other hand, SRN-GAN, by virtue of its softer constraints on the Lipschitz constant, trades off a higher Lipschitz constant for a better inception score—highlighting the flexibility provided by SRN. Additional experiments in Appendix E.2 show more detailed behaviour of GANs in regards to empirical lipschitz in a variety of settings.

| Model | IS | FID |
|---|---|---|
| SN-GAN | **9.04** | 23.2 |
| SRN-GAN (Our) | 8.85 | **19.55** |

Table 3: CIFAR100 experiments.

| Model | CIFAR10 | CelebA |
|---|---|---|
| SN-GAN | 10.69 | 0.36 |
| SRN-GAN (Our) | **11.97** | **0.64** |

Table 4: Neural Discriminator Loss (Higher the better).

## 6 CONCLUSION

We proposed a new normalization (SRN) that allows us to constrain the stable rank of each affine layer of a NN, which in turn learns a mapping with low empirical Lipschitz constant. We also provide optimality guarantees of SRN. On a variety of neural network architectures, we showed that SRN improves the generalization and memorization properties of a standard classifier. In addition, we show that SRN improves the training of GANs and provide better inception, FID, and ND scores.

## 7 ACKNOWLEDGEMENTS

The authors would like to thank Leonard Berrada and Pawan Kumar for helpful discussions. AS acknowledges support from The Alan Turing Institute under the Turing Doctoral Studentship grant TU/C/000023. PHS and PD are supported by the ERC grant ERC-2012-AdG 321162-HELIOS, EPSRC grant Seebibyte EP/M013774/1 and EPSRC/MURI grant EP/N019474/1. PHS and PD also acknowledges the Royal Academy of Engineering and FiveAI.

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

# A TECHNICAL PROOFS

Here we provide an extensive proof of Theorem 1 (Appendix A.1). We also provide the optimal solution to the spectral norm problem in Appendix A.2. Auxiliary lemmas on which our proof depends are provided in Appendix A.3.

## A.1 PROOF FOR OPTIMAL STABLE RANK NORMALIZATION. (MAIN THEOREM)

**Theorem 1.** *Given a real matrix $\mathbf{W} \in \mathbb{R}^{m \times n}$ with rank $p$, a target spectrum (or singular value) preservation index $k$ $(0 \leq k < p)$, and a target stable rank of $r$ $(1 \leq r < \mathrm{srank}(\mathbf{W}))$, the optimal solution $\widehat{\mathbf{W}}_k$ to problem (4) is $\widehat{\mathbf{W}}_k = \gamma_1 \mathbf{S}_1 + \gamma_2 \mathbf{S}_2$, where $\mathbf{S}_1 = \sum_{i=1}^{\max(1,k)} \sigma_i \mathbf{u}_i \mathbf{v}_i^\top$ and $\mathbf{S}_2 = \mathbf{W} - \mathbf{S}_1$. $\{\sigma_i\}_{i=1}^k$, $\{\mathbf{u}_i\}_{i=1}^k$ and $\{\mathbf{v}_i\}_{i=1}^k$ are the top $k$ singular values and vectors of $\mathbf{W}$, and, depending on $k$, $\gamma_1$ and $\gamma_2$ are defined below. For simplicity, we first define $\gamma = \frac{\sqrt{r\sigma_1^2 - \|\mathbf{S}_1\|_F^2}}{\|\mathbf{S}_2\|_F}$, then*

a) *If $k = 0$ (no spectrum preservation), the problem becomes non-convex, the optimal solution to which is obtained for $\gamma_2 = \dfrac{\gamma + r - 1}{r}$ and $\gamma_1 = \dfrac{\gamma_2}{\gamma}$, when $r > 1$. If $r = 1$, then $\gamma_2 = 0$ and $\gamma_1 = 1$. Since, in this case, $\|\mathbf{S}_1\|_F^2 = \sigma_1^2$, $\gamma = \dfrac{\sqrt{r-1}\sigma_1}{\|\mathbf{S}_2\|_F}$.*

b) *If $k \geq 1$, the problem is convex. If $r \geq \dfrac{\|\mathbf{S}_1\|_F^2}{\sigma_1^2}$ the optimal solution is obtained for $\gamma_1 = 1$, and $\gamma_2 = \gamma$ and if not, the problem is not feasible.*

c) *Also, $\left\| \widehat{\mathbf{W}}_k - \mathbf{W} \right\|_F$ monotonically increases with $k$ for $k \geq 1$.*

*Proof.* Here we provide the proof of Theorem 1 (in the main paper) for all the three cases with optimality and uniqueness guarantees. Let $\widehat{\mathbf{W}}_k$ be the optimal solution to the problem for any of the two cases. From Lemma 5, the SVD of $\mathbf{W}$ and $\widehat{\mathbf{W}}_k$ can be written as $\mathbf{W} = \mathbf{U}\Sigma\mathbf{V}^\top$ and $\widehat{\mathbf{W}}_k = \mathbf{U}\Lambda\mathbf{V}^\top$, respectively. Then, $L = \left\| \mathbf{W} - \widehat{\mathbf{W}}_k \right\|_F^2 = \langle \Sigma - \Lambda, \Sigma - \Lambda \rangle_F$. From now onwards, we denote $\Sigma$ and $\Lambda$ as vectors consisting of the diagonal entries, and $\langle ., . \rangle$ as the vector inner product [10].

**Proof for Case (a):** In this case, there is no constraint enforced to preserve any of the singular values of the given matrix while obtaining the new one. The only constraint is that the new matrix should have the stable rank of $r$. Let us assume $\Sigma = (\sigma_1, \cdots, \sigma_p)$, $\Sigma_2 = (\sigma_2, \cdots, \sigma_p)$, $\Lambda = (\lambda_1, \cdots, \lambda_p)$ and $\Lambda_2 = (\lambda_2, \cdots, \lambda_p)$. Using these notations, we can write $L$ as:

$$
\begin{aligned}
L &= \langle \Sigma, \Sigma \rangle + \langle \Lambda, \Lambda \rangle - 2 \langle \Sigma, \Lambda \rangle \\
&= \langle \Sigma, \Sigma \rangle + \lambda_1^2 + \langle \Lambda_2, \Lambda_2 \rangle - 2\sigma_1\lambda_1 - 2 \langle \Sigma_2, \Lambda_2 \rangle
\end{aligned}
\tag{6}
$$

Using the stable rank constraint $\mathrm{srank}(\widehat{\mathbf{W}}_k) = r$, which is $r = 1 + \dfrac{\sum_{j=2}^p \lambda_j^2}{\lambda_1^2}$.

**Case for r > 1** If $r > 1$ we obtain the following equality constraint, making the problem non-convex.

$$
\lambda_1^2 = \frac{\langle \Lambda_2, \Lambda_2 \rangle}{r - 1}
\tag{7}
$$

However, we will show that the solution we obtain is optimal and unique. Substituting (7) into (6) we get

$$
L = \langle \Sigma, \Sigma \rangle + \frac{\langle \Lambda_2, \Lambda_2 \rangle}{r - 1} + \langle \Lambda_2, \Lambda_2 \rangle - 2\sigma_1 \sqrt{\frac{\langle \Lambda_2, \Lambda_2 \rangle}{r - 1}} - 2 \langle \Sigma_2, \Lambda_2 \rangle
\tag{8}
$$

---

[10] $\langle ., . \rangle_F$ represents the Frobenius inner product of two matrices, which in the case of diagonal matrices is the same as the inner product of the diagonal vectors.

Setting $\frac{\partial L}{\partial \Lambda_2} = 0$ to get the family of critical points

$$\frac{2\Lambda_2}{r-1} + 2\Lambda_2 - \frac{4\sigma_1 \Lambda_2}{2\sqrt{(r-1)\langle \Lambda_2, \Lambda_2 \rangle}} - 2\Sigma_2 = 0$$

$$\implies \Sigma_2 = \Lambda_2 \left( \frac{1}{r-1} + 1 - \frac{\sigma_1}{1\sqrt{(r-1)\langle \Lambda_2, \Lambda_2 \rangle}} \right) \quad (9)$$

$$\implies \frac{\Sigma_2[i]}{\lambda_2[i]} = \left( \frac{1}{r-1} + 1 - \frac{\sigma_1}{1\sqrt{(r-1)\langle \Lambda_2, \Lambda_2 \rangle}} \right) = \frac{1}{\gamma_2} \quad \forall\, 1 \le i \le p \quad (10)$$

As the R.H.S. of 10 is independant of $i$, the above equality implies that all the critical points of (8) are a scalar multiple of $\Sigma_2$, implying, $\Lambda_2 = \gamma_2 \Sigma_2$. Note that the domain of $\Lambda_2$ are all strictly positive vectors and thus, we can ignore the critical point at $\Lambda_2 = \mathbf{0}$. Substituting this into (9) we obtain

$$\Sigma_2 = \gamma_2 \Sigma_2 \left( \frac{1}{r-1} + 1 - \frac{\sigma_1}{\gamma_2 \sqrt{(r-1)\langle \Sigma_2, \Sigma_2 \rangle}} \right)$$

Using the fact that $\langle \Sigma_2, \Sigma_2 \rangle = \|\mathbf{S}_2\|_F^2$ in the above equality and with some algebraic manipulations, we obtain $\gamma_2 = \frac{\gamma+r-1}{r}$ where, $\gamma = \frac{\sqrt{r-1}\sigma_1}{\|\mathbf{S}_2\|_F}$. Note, $r \ge 1$, $\gamma \ge 0$, and $\Sigma \ge 0$, implying, $\Lambda_2 = \gamma_2 \Sigma_2 \ge 0$.

**Local minima:** Now, we will show that $\Lambda_2$ is indeed a minima of (8). To show this, we compute the hessian of $L$. Recall that

$$\frac{\partial L}{\partial \Lambda} = \frac{2r}{r-1}\Lambda - \frac{2\sigma_1 \Lambda}{\sqrt{(r-1)\|\Lambda\|_2^2}} - 2\Sigma_2$$

$$\mathbf{H} = \frac{\partial^2 L}{\partial^2 \Lambda} = \frac{2r}{r-1}\mathbf{I} - \frac{2\sigma_1}{\sqrt{(r-1)\|\Lambda\|_2^2}} \left( \|\Lambda\|_2 \mathbf{I} - \frac{1}{\|\Lambda\|_2}\Lambda\Lambda^\top \right)$$

$$= 2\left( \frac{r}{r-1} - \frac{\sigma_1 \|\Lambda\|_2}{\sqrt{(r-1)\|\Lambda\|_2^2}} \right)\mathbf{I} + \frac{2\sigma_1}{\sqrt{r-1}\|\Lambda\|_2^3}\left( \Lambda\Lambda^\top \right)$$

Now we need to show that $\mathbf{H}$ at the solution $\Lambda_2$ is PSD i.e. $\forall\, \mathbf{x} \in \mathbb{R}^{p-1}$, $\mathbf{x}^\top \mathbf{H}(\mathbf{\Lambda_2})\mathbf{x} \ge 0$

$$\mathbf{x}^\top \mathbf{H}\mathbf{x} = 2\left( \frac{r}{r-1} - \frac{\sigma_1 \|\Lambda\|_2}{\sqrt{(r-1)}\|\Lambda\|_2^2} \right)\|\mathbf{x}_2^2\| + \frac{2\sigma_1}{\sqrt{r-1}\|\Lambda\|_2^3}\mathbf{x}^\top \left( \Lambda\Lambda^\top \right)\mathbf{x}$$

$$\overset{(a)}{\ge} 2\left( \frac{r}{r-1} - \frac{\sigma_1 \|\Lambda\|_2}{\sqrt{(r-1)}\|\Lambda\|_2^2} \right)\|\mathbf{x}\|_2^2$$

$$\overset{(b)}{\ge} 2\left( \frac{r}{r-1} - \frac{\sigma_1}{(r-1)\lambda_1} \right)\|\mathbf{x}\|_2^2 \overset{(c)}{=} \frac{2r}{r-1}\left( 1 - \frac{\gamma}{(\gamma+r-1)} \right)\|\mathbf{x}\|_2^2$$

$$\overset{(d)}{\ge} \frac{2r}{r-1}\left( 1 - \frac{1}{(1+r-1)} \right)\|\mathbf{x}\|_2^2 = 2\|\mathbf{x}\|_2^2 \ge 0$$

Here $(a)$ is due to the fact that the matrix $\Lambda\Lambda^\top$ is an outer product matrix and is hence PSD. $(b)$ follows due to (7) and $(c)$ follows by substituting $\lambda_1 = \gamma_1 \sigma_1$ and then the value of $\gamma_1$. Finally $(d)$ follows as $\left( 1 - \frac{\gamma}{(\gamma+r-1)} \right)$ is decreasing with respect to $\gamma$ and we know that $\gamma < 1$ due to the assumption that $\mathrm{srank}(\mathbf{W}) < r$. Thus, we can substitute $\gamma = 1$ to find the minimum value of the expression. This concludes our proof that $\Lambda_2$ is indeed a local minima of $L$.

**Uniqueness:** The uniqueness of $\Lambda_2$ as a solution to (8) is shown in Lemma 6 and is also guaranteed by the fact that $\gamma_2$ has a unique value. Using $\Lambda_2 = \gamma_2 \Sigma_2$ and $\lambda_1 = \gamma_1 \sigma_1$ in (7), we obtain a unique solution $\gamma_1 = \frac{\gamma_2}{\gamma}$.

Now, we need to show that it is also an unique solution to Theorem 1.

For all solutions to Theorem 1 that have singular vectors which are different than that of $\mathbf{W}$, by Lemma 5, the matrix formed by replacing the singular vectors of the solution with that of $\mathbf{W}$ is also a solution. Thus, if there were a solution with different singular values than $\widehat{\mathbf{W}}_k$, it should have appeared as a solution to (8). However, we have shown that (8) has a unique solution.

Now, we need to show that among all matrices with the same singular values as that of $\widehat{\mathbf{W}}_k$, $\widehat{\mathbf{W}}_k$ is strictly better in terms of $\left\|\mathbf{W} - \widehat{\mathbf{W}}_k\right\|$. This requires a further assumption that every non-zero singular value of $\Lambda_2$ has a multiplicity of 1 i.e. they are all distinctly unique. Intuitively, this doesn't allow to create a different matrix by simply interchanging the singular vectors associated with the equal singular values. As the elements of $\Sigma_2$ are distinct, the elements of $\Lambda_2 = \gamma_2 \Sigma_2$ are also distinct and thus by the second part of Lemma 5, $\widehat{\mathbf{W}}_k$ is strictly better, in terms of $\left\|\mathbf{W} - \widehat{\mathbf{W}}_k\right\|$, than all matrices which have the same singular values as that of $\widehat{\mathbf{W}}_k$. This concludes our discussion on the uniqueness of the solution.

**Case for r = 1:** Substituting $r = 1$ in the constraint $r = 1 + \frac{\sum_{j=2}^p \lambda_j^2}{\lambda_1^2}$ we get

$$r - 1 = \frac{\sum_{j=2}^p \lambda_j^2}{\lambda_1^2} = 0 \implies \sum_{j=2}^p \lambda_j^2 = 0$$

As it is a sum of squares, each of the individual elements is also zero i.e. $\lambda_j = 0 \; \forall 2 \leq j \leq p$. Substituting this into (6), we get the following quadratic equation in $\lambda_1$

$$L = \langle \Sigma, \Sigma \rangle + \lambda_1^2 - 2\sigma_1 \lambda_1 \tag{11}$$

which is minimized at $\lambda_1 = \sigma_1$, thus proving that $\gamma_1 = 1$ and $\gamma_2 = 0$.

**Proof for Case (b):** In this case, the constraints are meant to preserve the top $k$ singular values of the given matrix while obtaining the new one. Let $\Sigma_1 = (\sigma_1, \cdots, \sigma_k)$, $\Sigma_2 = (\sigma_{k+1}, \cdots, \sigma_p)$, $\Lambda_1 = (\lambda_1, \cdots, \lambda_k)$, $\Lambda_2 = (\lambda_{k+1}, \cdots, \lambda_p)$. Since satisfying all the constraints imply $\Sigma_1 = \Lambda_1$, thus, $L := \left\|\mathbf{W} - \widehat{\mathbf{W}}_k\right\|_{\text{F}}^2 = \langle \Sigma_2 - \Lambda_2, \Sigma_2 - \Lambda_2 \rangle$. From the stable rank constraint $\text{srank}(\widehat{\mathbf{W}}_k) = r$, we have

$$r = \frac{\langle \Lambda_1, \Lambda_1 \rangle + \langle \Lambda_2, \Lambda_2 \rangle}{\lambda_1^2}$$

$$\therefore \; \langle \Lambda_2, \Lambda_2 \rangle = r\lambda_1^2 - \langle \Lambda_1, \Lambda_1 \rangle = r\sigma_1^2 - \langle \Sigma_1, \Sigma_1 \rangle \tag{12}$$

The above equality constraint makes the problem non-convex. Thus, we relax it to $\text{srank}(\widehat{\mathbf{W}}_k) \leq r$ to make it a convex problem and show that the optimality is achieved with equality. Let $r\sigma_1^2 - \langle \Sigma_1, \Sigma_1 \rangle = \eta$. Then, the relaxed problem can be written as

$$\min_{\Lambda_2 \in \mathbb{R}^{p-k}} L := \langle \Sigma_2 - \Lambda_2, \Sigma_2 - \Lambda_2 \rangle$$

$$\text{s.t.} \quad \Lambda_2 \geq 0, \langle \Lambda_2, \Lambda_2 \rangle \leq \eta.$$

We introduce the Lagrangian dual variables $\Gamma \in \mathbb{R}^{p-k}$ and $\mu$ corresponding to the positivity and the stable rank constraints, respectively. The Lagrangian can then be written as

$$\mathcal{L}(\Lambda_2, \Gamma, \mu)_{\Gamma \geq \mathbf{0}, \mu \geq 0} = \langle \Sigma_2 - \Lambda_2, \Sigma_2 - \Lambda_2 \rangle + \mu (\langle \Lambda_2, \Lambda_2 \rangle - \eta) - \langle \Gamma, \Lambda_2 \rangle \tag{13}$$

Using the primal optimality condition $\frac{\partial \mathcal{L}}{\partial \Lambda_2} = \mathbf{0}$, we obtain

$$2\Lambda_2 - 2\Sigma_2 + 2\mu\Lambda_2 - \Gamma = \mathbf{0}$$

$$\implies \Lambda_2 = \frac{\Gamma + 2\Sigma_2}{2(1 + \mu)} \tag{14}$$

Using the above condition on $\Lambda_2$ with the constraint $\langle \Lambda_2, \Lambda_2 \rangle \leq \eta$, combined with the stable rank constraint of the given matrix $\mathbf{W}$ that comes with the problem definition, $\mathrm{srank}(\mathbf{W}) > r$ (which implies $\langle \Sigma_2, \Sigma_2 \rangle > \eta$), the following inequality must be satisfied for any $\Gamma \geq 0$

$$1 < \frac{\langle \Sigma_2, \Sigma_2 \rangle}{\eta} \leq \frac{\langle \Gamma + \Sigma_2, \Gamma + \Sigma_2 \rangle}{\eta} \leq (1 + \mu)^2 \tag{15}$$

For the above inequality to satisfy, the dual variable $\mu$ must be greater than zero, implying, $\langle \Lambda_2, \Lambda_2 \rangle - \eta$ must be zero for the complementary slackness to satisfy. Using this with the optimality condition (14) we obtain

$$(1 + \mu)^2 = \frac{\langle \Gamma + 2\Sigma_2, \Gamma + 2\Sigma_2 \rangle}{4\eta}$$

Substituting the above solution back into the primal optimality condition we get

$$\Lambda_2 = (\Gamma + 2\Sigma_2) \frac{\sqrt{\eta}}{\sqrt{\langle \Gamma + 2\Sigma_2, \Gamma + 2\Sigma_2 \rangle}} \tag{16}$$

Finally, we use the complimentary slackness condition $\Gamma \odot \Lambda_2 = \mathbf{0}$[11] to get rid of the dual variable $\Gamma$ as follows

$$\Gamma \odot (\Gamma + 2\Sigma_2) \frac{\sqrt{\eta}}{\sqrt{\langle \Gamma + 2\Sigma_2, \Gamma + 2\Sigma_2 \rangle}} = \mathbf{0}$$

It is easy to see that the above condition is satisfied only when $\Gamma = \mathbf{0}$ as $\Sigma_2 \geq \mathbf{0}$ and $\eta > 0$. Therefore, using $\Gamma = \mathbf{0}$ in (16) we obtain the optimal solution of $\Lambda_2$ as

$$\Lambda_2 = \frac{\sqrt{\eta}}{\sqrt{\langle \Sigma_2, \Sigma_2 \rangle}} \Sigma_2 = \frac{\sqrt{r\sigma_1^2 - \|\mathbf{S}_1\|_F^2}}{\|\mathbf{S}_2\|_F^2} \Sigma_2 = \gamma \Sigma_2 \tag{17}$$

**Proof for Case (c):** The monotonicity of $\left\|\widehat{\mathbf{W}}_k - \mathbf{W}\right\|_F$ for $k \geq 1$ is shown in Lemma 3. $\qquad \square$

Note that by the assumption that $\mathrm{srank}(\mathbf{W}) < r$, we can say that $\gamma < 1$. Therefore in all the cases $\gamma_2 < 1$. Let us look at the required conditions for $\gamma_1 \geq 1$ to hold. When $k \geq 1$, $\gamma_1 = 1$ holds. When $k = 0$, for $\gamma_1 > 1$ to be true, $\gamma_2 < \gamma$ should hold, implying, $(\gamma - 1) < r(\gamma - 1)$, which is always true as $r > 1$ (by the definition of stable rank).

**Lemma 3.** *For $k \geq 1$, the solution to the optimization problem (4) obtained using Theorem 1 is closest to the original matrix $\mathbf{W}$ in terms of Frobenius norm when only the spectral norm is preserved, implying, $k = 1$.*

*Proof.* For a given matrix $\mathbf{W}$ and a partitioning index $k \in \{1, \cdots, p\}$, let $\widehat{\mathbf{W}}_k = \mathbf{S}_1^k + \gamma \mathbf{S}_2^k$ be the matrix obtained using Theorem 1. We use the superscript $k$ along with $\mathbf{S}_1$ and $\mathbf{S}_2$ to denote that this refers to the particular solution of $\widehat{\mathbf{W}}_k$. Plugging the value of $\gamma$ and using the fact that $\left\|\mathbf{S}_2^k\right\|_F \neq 0$, we can write

$$\left\|\mathbf{W} - \widehat{\mathbf{W}}_k\right\|_F = (1 - \gamma) \left\|\mathbf{S}_2^k\right\|_F$$

$$= \left\|\mathbf{S}_2^k\right\|_F - \sqrt{r\sigma_1^2 - \left\|\mathbf{S}_1^k\right\|_F^2}$$

$$= \left\|\mathbf{S}_2^k\right\|_F - \sqrt{r\sigma_1^2 - \|\mathbf{W}\|_F^2 + \left\|\mathbf{S}_2^k\right\|_F^2}.$$

Thus, $\left\|\mathbf{W} - \widehat{\mathbf{W}}_k\right\|_F$ can be written in a simplified form as $f(x) = x - \sqrt{a + x^2}$, where $x = \left\|\mathbf{S}_2^k\right\|_F$ and $a = r\sigma_1^2 - \|\mathbf{W}\|_F^2$. Note, $a \leq 0$ as $1 \leq r \leq \mathrm{srank}(\mathbf{W})$, and $a + x^2 \geq 0$ because of the condition in Theorem 1. Under these settings, it is trivial to verify that $f$ is a monotonically decreasing function of $x$. Using the fact that as the partition index $k$ increases, $x$ decreases, it is straightforward to conclude that the minimum of $f(x)$ is obtained at $k = 1$. $\qquad \square$

---

[11] $\odot$ is the hadamard product

## A.2 PROOF FOR OPTIMAL SPECTRAL NORMALIZATION

The widely used spectral normalization (Miyato et al., 2018) where the given matrix $\mathbf{W} \in \mathbb{R}^{m \times n}$ is divided by the maximum singular value is an approximation to the optimal solution of the spectral normalization problem defined as

$$\underset{\widehat{\mathbf{W}}}{\arg\min} \left\| \mathbf{W} - \widehat{\mathbf{W}} \right\|_{\mathrm{F}}^{2} \tag{18}$$

$$s.t. \quad \sigma(\widehat{\mathbf{W}}) \leq s,$$

where $\sigma(\widehat{\mathbf{W}})$ denotes the maximum singular value and $s > 0$ is a hyperparameter. The optimal solution to this problem is shown in Algorithm 3. In what follows we provide the optimality proof

---

**Algorithm 3** Spectral Normalization

---

**Require:** $\mathbf{W} \in \mathbb{R}^{m \times n}$, $s$
1: $\mathbf{W}_1 \leftarrow \mathbf{0}$, $p \leftarrow \min(m, n)$
2: **for** $k \in \{1, \cdots, p\}$ **do**
3:      $\{\mathbf{u}_k, \mathbf{v}_k, \sigma_k\} \leftarrow SVD(\mathbf{W}, k)$          $\triangleright$ perform power method to get $k$-th singular value
4:      **if** $\sigma_k \geq s$ **then**
5:          $\mathbf{W}_1 \leftarrow \mathbf{W}_1 + s\, \mathbf{u}_k \mathbf{v}_k^{\top}$
6:          $\mathbf{W} \leftarrow \mathbf{W} - \sigma_k\, \mathbf{u}_k \mathbf{v}_k^{\top}$
7:      **else**
8:          break          $\triangleright$ exit for loop
9:      **end if**
10: **end for**
11: **return** $\mathbf{W} \leftarrow \mathbf{W}_1 + \mathbf{W}$

---

of Algorithm 3 for the sake of completeness. Let $\mathrm{SVD}(\mathbf{W}) = \mathbf{U}\Sigma\mathbf{V}^{\top}$ and let us assume that $\mathbf{Z} = \mathbf{S}\Lambda\mathbf{T}^{\top}$ is a solution to the problem 18. Trivially, $\mathbf{X} = \mathbf{U}\Lambda\mathbf{V}^{\top}$ also satisfies $\sigma(\mathbf{X}) \leq s$. Now, $\|\mathbf{W} - \mathbf{X}\|_{\mathrm{F}}^{2} = \left\| \mathbf{U}\left(\Sigma - \Lambda\right)\mathbf{V}^{\top} \right\|_{\mathrm{F}}^{2} = \|(\Sigma - \Lambda)\|_{\mathrm{F}}^{2} \leq \|\mathbf{W} - \mathbf{Z}\|_{\mathrm{F}}^{2}$, where the last inequality directly comes from Lemma 4. Thus the singular vectors of the optimal solution must be the same as that of $\mathbf{W}$. This boils down to solving the following problem

$$\underset{\Lambda \in \mathbb{R}_{+}^{\min(m,n)}}{\arg\min} \|\Lambda - \Sigma\|_{\mathrm{F}}^{2} \ \ s.t. \ \Lambda[i] \leq s \ \forall i \in \{0, \min(m, n)\}. \tag{19}$$

Here, without loss of generality, we abuse notations by considering $\Lambda$ and $\Sigma$ to represent the diagonal vectors of the original diagonal matrices $\Lambda$ and $\Sigma$, and $\Lambda[i]$ as its $i$-th index. It is trivial to see that the optimal solution with minimum Frobenius norm is achieved when

$$\Lambda[i] = \begin{cases} \Sigma[i], & \text{if } \Sigma[i] \leq s \\ s, & \text{otherwise.} \end{cases}$$

This is exactly what Algorithm 3 implements.

## A.3 AUXILIARY LEMMAS

**Lemma 4.** *[Reproduced from Theorem 5 in Mirsky (1960)] For any two matrices* $\mathbf{A}, \mathbf{B} \in \mathbb{R}^{m \times n}$ *with singular values as* $\sigma_1 \geq \cdots \geq \sigma_n$ *and* $\rho_1 \geq \cdots \geq \rho_n$, *respectively*

$$\|\mathbf{A} - \mathbf{B}\|_{\mathrm{F}}^{2} \geq \sum_{i=1}^{n} (\sigma_i - \rho_i)^2$$

*Proof.* Consider the following symmetric matrices

$$\mathbf{X} = \begin{bmatrix} \mathbf{0} & \mathbf{A} \\ \mathbf{A}^{\top} & \mathbf{0} \end{bmatrix}, \mathbf{Y} = \begin{bmatrix} \mathbf{0} & \mathbf{B} \\ \mathbf{B}^{\top} & \mathbf{0} \end{bmatrix}, \mathbf{Z} = \begin{bmatrix} \mathbf{0} & \mathbf{A} - \mathbf{B} \\ (\mathbf{A} - \mathbf{B})^{\top} & \mathbf{0} \end{bmatrix}$$

Let $\tau_1 \geq \cdots \geq \tau_n$ be the singular values of $\mathbf{Z}$. Then the set of characteristic roots of $\mathbf{X}, \mathbf{Y}$ and $\mathbf{Z}$ in descending order are $\{\rho_1, \cdots, \rho_n, -\rho_n, \cdots, -\rho_1\}$, $\{\sigma_1, \cdots, \sigma_n, -\sigma_n, \cdots, -\sigma_1\}$, and $\{\tau_1, \cdots, \tau_n, -\tau_n, \cdots, -\tau_1\}$, respectively. By Lemma 2 in Wielandt (1955)

$$[\sigma_1 - \rho_1, \cdots, \sigma_n - \rho_n, \rho_n - \sigma_n, \cdots, \rho_1 - \sigma_1] \preceq [\tau_1, \cdots \tau_n, -\tau_n, -\tau_1],$$

which implies that

$$\sum_{i=1}^{n} (\sigma_i - \rho_i)^2 \leq \sum_{i=1}^{n} \tau_i^2 = \|\mathbf{A} - \mathbf{B}\|_{\mathrm{F}}^2 \qquad (20)$$

$\square$

**Lemma 5.** *Let* $\mathbf{A}, \mathbf{B} \in \mathbb{R}^{m \times n}$ *where* $\mathrm{SVD}(\mathbf{A}) = \mathbf{U}\Sigma\mathbf{V}^\top$ *and* $\mathbf{B}$ *is the solution to the following problem*

$$\mathbf{B} = \arg\min_{\mathrm{srank}(\mathbf{W}) = r} \|\mathbf{W} - \mathbf{A}\|_{\mathrm{F}}^2. \qquad (21)$$

*Then,* $\mathrm{SVD}(\mathbf{B}) = \mathbf{U}\Lambda\mathbf{V}^\top$ *where* $\Lambda$ *is a diagonal matrix with non-negative entries. Implying,* $\mathbf{A}$ *and* $\mathbf{B}$ *will have the same singular vectors.*

*Proof.* Let us assume that $\mathbf{Z} = \mathbf{S}\Lambda\mathbf{T}^\top$ is a solution to the problem 21 where $\mathbf{S} \neq \mathbf{U}$ and $\mathbf{T} \neq \mathbf{V}$. Trivially, $\mathbf{X} = \mathbf{U}\Lambda\mathbf{V}^\top$ also lies in the feasible set as it satisfies $\mathrm{srank}(\mathbf{X}) = r$ (note stable rank only depends on the singular values). Using the fact that the Frobenius norm is invariant to unitary transformations, we can write $\|\mathbf{A} - \mathbf{X}\|_{\mathrm{F}}^2 = \left\|\mathbf{U}(\Sigma - \Lambda)\mathbf{V}^\top\right\|_{\mathrm{F}}^2 = \|(\Sigma - \Lambda)\|_{\mathrm{F}}^2$. Combining this with Lemma 4, we obtain $\|\mathbf{A} - \mathbf{X}\|_{\mathrm{F}}^2 = \|(\Sigma - \Lambda)\|_{\mathrm{F}}^2 \leq \|\mathbf{A} - \mathbf{Z}\|_{\mathrm{F}}^2$. Since, $\mathbf{S} \neq \mathbf{U}$ and $\mathbf{T} \neq \mathbf{V}$, we can further change $\leq$ to a strict inequality $<$. This completes the proof.

Generally speaking, the optimal solution to problem 21 with constraints depending only on the singular values (*e.g.* stable rank in this case) will have the same singular vectors as that of the original matrix.

Further the inequality in (20) can be converted into a strict inequality if neither of $\mathbf{A}$ and $\mathbf{B}$ have repeated singular values. Using that strict inequality, if both $\Sigma$ and $\Lambda$ have no repeated values, then $\mathbf{B}$ is the only solution to (21) that has the singular values of $\Lambda$.

$\square$

**Lemma 6.** *Let* $\mathbf{y}_1 = a\mathbf{x}_1 + b\hat{\mathbf{x}}_1$ *and* $\mathbf{y}_2 = a\mathbf{x}_2 + b\hat{\mathbf{x}}_2$, *where* $\hat{\mathbf{x}}_1$ *and* $\hat{\mathbf{x}}_2$ *denotes the unit vectors. Then,* $\mathbf{y}_1 = \mathbf{y}_2$ *if* $\mathbf{x}_1 = \mathbf{x}_2$.

# B  EMPIRICAL LIPSCHITZ CONSTANT

## B.1  RELATING EMPIRICAL LOCAL AND GLOBAL LIPSCHITZ CONSTANTS

**Proposition B.1.** *Let* $f : \mathbb{R}^m \mapsto \mathbb{R}$ *be a Fréchet differentiable function,* $\mathcal{D}$ *the dataset, and* $\mathrm{Conv}(\mathbf{x}_i, \mathbf{x}_j)$ *denotes the convex combination of a pair of samples* $\mathbf{x}_i$ *and* $\mathbf{x}_j$, *then* $\forall p, q \in [1, \infty]$ *such that* $\frac{1}{p} + \frac{1}{q} = 1$

$$\max_{\mathbf{x}_i, \mathbf{x}_j \in \mathcal{D}} \frac{|f(\mathbf{x}_i) - f(\mathbf{x}_j)|}{\|\mathbf{x}_i - \mathbf{x}_j\|_p} \leq \max_{\substack{\mathbf{x}_i, \mathbf{x}_j \in \mathcal{D} \\ \mathbf{x} \in \mathrm{Conv}(\mathbf{x}_i, \mathbf{x}_j)}} \|J_f(\mathbf{x})\|_q$$

*Proof.* Let $f : \mathbb{R}^m \to \mathbb{R}$ be a differentiable function on an open set containing $\mathbf{x}_i$ and $\mathbf{x}_j$ such that $\mathbf{x}_i \neq \mathbf{x}_j$. By applying fundamental theorem of calculus

$$
\begin{aligned}
|f(\mathbf{x}_i) - f(\mathbf{x}_j)| &= \left| \int_0^1 \nabla f(\mathbf{x}_i + \theta(\mathbf{x}_j - \mathbf{x}_i))^\top (\mathbf{x}_j - \mathbf{x}_i) \, \partial\theta \right| \\
&\leq \int_0^1 \left| \nabla f(\mathbf{x}_i + \theta(\mathbf{x}_j - \mathbf{x}_i))^\top (\mathbf{x}_j - \mathbf{x}_i) \right| \partial\theta \\
&\overset{(a)}{\leq} \int_0^1 \|\nabla f(\mathbf{x}_i + \theta(\mathbf{x}_j - \mathbf{x}_i))\|_q \|(\mathbf{x}_j - \mathbf{x}_i)\|_p \, \partial\theta \\
&\leq \int_0^1 \max_{\theta \in (0,1)} \|\nabla f(\mathbf{x}_i + \theta(\mathbf{x}_j - \mathbf{x}_i))\|_q \|(\mathbf{x}_j - \mathbf{x}_i)\|_p \, \partial\theta \\
&= \max_{\theta \in (0,1)} \|\nabla f(\mathbf{x}_i + \theta(\mathbf{x}_j - \mathbf{x}_i))\|_q \|(\mathbf{x}_j - \mathbf{x}_i)\|_p \int_0^1 \partial\theta
\end{aligned}
$$

$$
\therefore \frac{|f(\mathbf{x}_i) - f(\mathbf{x}_j)|}{\|(\mathbf{x}_j - \mathbf{x}_i)\|_p} \leq \max_{\theta \in (0,1)} \|\nabla f(\mathbf{x}_i + \theta(\mathbf{x}_j - \mathbf{x}_i))\|_q = \max_{\mathbf{x} \in Conv(\mathbf{x}_i, \mathbf{x}_j)} \|\nabla f(\mathbf{x})\|_q.
$$

The inequality (a) is due to Hölder's inequality. $\square$

## B.2 Effect of Rank on the Empirical Lipschitz Constants

Let $f(\mathbf{x}) = \mathbf{W}_2 \mathbf{W}_1 \mathbf{x}$ be a two-layer linear NN with weights $\mathbf{W}_1$ and $\mathbf{W}_2$. The Jacobian in this case is independent of $\mathbf{x}$. Thus, the local Lipschitz constant is the same for all $\mathbf{x} \in \mathbb{R}^m$, implying, local $L_e = L_l(\mathbf{x}) = L_l = \|\mathbf{W}_2 \mathbf{W}_1\| \leq \|\mathbf{W}_2\| \|\mathbf{W}_1\|$. Note, in the case of 2-matrix norm reducing the rank will not affect the upperbound. However, as will be discussed below, rank reduction greatly influences the global $L_e$.

Let $\mathbf{x}_i$ and $\mathbf{x}_j$ be random pairs from $\mathcal{D}$ and $\Delta\mathbf{x} \neq \mathbf{0}$ be the difference $\mathbf{x}_i - \mathbf{x}_j$, then, the global $L_e$ is $\max_{\{\mathbf{x}_i, \mathbf{x}_j\} \in \mathcal{D}} \frac{\|\mathbf{W}_2 \mathbf{W}_1 \Delta\mathbf{x}\|}{\|\Delta\mathbf{x}\|}$. Let $k_1$ and $k_2$ be the ranks, and $\sigma_1 \geq \cdots \geq \sigma_{k_1}$ and $\lambda_1 \geq \cdots \geq \lambda_{k_2}$ the singular values of the matrices $\mathbf{W}_1$ and $\mathbf{W}_2$, respectively. Let $P_i = \mathbf{u}_i \bar{\mathbf{u}}_i^\top$ be the orthogonal projection matrix corresponding to $\mathbf{u}_i$ and $\bar{\mathbf{u}}_i$, the left and the right singular vectors of $\mathbf{W}_1$. Similarly, we define $Q_i$ for $\mathbf{W}_2$ corresponding to $\mathbf{v}_i$ and $\bar{\mathbf{v}}_i$. Then, $\mathbf{W}_2 \mathbf{W}_1 = \sum_{i=1}^{k_2} \sum_{j=1}^{k_1} \lambda_i \sigma_j Q_i P_j$. The upperbound, $\lambda_1 \sigma_1$, can be achieved if and only if $\Delta\mathbf{x} = \bar{\mathbf{u}}_1 \|\Delta\mathbf{x}\|$ and $\mathbf{u}_1 = \bar{\mathbf{v}}_1$ (a perfect alignment), which is highly unlikely. In practice, not just the maximum singular values, as is the case with the Lipschitz upper-bound, rather the combination of the projection matrices and the singular values play a crucial role in providing an estimate of global $L_e$. Thus, reducing the singular values, which is equivalent to minimizing the rank (or stable rank), will directly affect $L_e$. For example, assigning $\sigma_j = 0$, which in effect will reduce the rank of $\mathbf{W}_1$ by one, will nullify its influence on all projections associated with $P_j$. Implying, all the $k_2$ projections $\sigma_j (\sum_{i=1}^{k_2} \lambda_i Q_i) P_j$ that would propagate the input via $P_j$ will be blocked. This, in effect, will influence $\|\mathbf{W}_2 \mathbf{W}_2 \Delta\mathbf{x}\|$; hence the global $L_e$. In a more general setting, let $k_i$ be the rank of the $i$-th linear layer, then, each singular value of a $j$-th layer can influence the maximum of $\prod_{i=1}^{j-1} k_i \prod_{i=j+1}^l k_i$ many paths through which an input can be propagated. Thus, mappings with low rank (stable) will greatly reduce the global $L_e$. Similar arguments can be drawn for local $L_e$ in the case of NN with non-linearity.

## C The local Lipschitz upper-bound for Neural Networks

As mentioned in Section 2, $L_l(\mathbf{x}) = \|J_f(\mathbf{x})\|_{p,q}$, where, in the case of NN, the Jacobian is:

$$
J_f(\mathbf{x}) = \frac{\partial f(\mathbf{x})}{\partial \mathbf{x}} := \frac{\partial \mathbf{z}_1}{\partial \mathbf{x}} \frac{\partial \phi_1(\mathbf{z}_1)}{\partial \mathbf{z}_1} \cdots \frac{\partial \mathbf{z}_l}{\partial \mathbf{a}_{l-1}} \frac{\partial \phi_l(\mathbf{z}_l)}{\partial \mathbf{z}_l}. \tag{22}
$$

Using $\frac{\partial \mathbf{z}_l}{\partial \mathbf{a}_{l-1}} = \mathbf{W}_l$ (affine transformation), and applying submultiplicativity of the matrix norms:

$$
\|J_f(\mathbf{x})\|_{p,q} \leq \|\mathbf{W}_1\|_{p,q} \left\| \frac{\partial \phi_1(\mathbf{z}_1)}{\partial \mathbf{z}_1} \right\| \cdots \|\mathbf{W}_l\|_{p,q} \left\| \frac{\partial \phi_l(\mathbf{z}_l)}{\partial \mathbf{z}_l} \right\|. \tag{23}
$$

Note, most commonly used activation functions $\phi(.)$ such as ReLU, sigmoid, tanh and maxout are known to have Lipschitz constant of 1 (if scaled appropriately)[12], thus, the upper bound can further be written only using the operator norms of the intermediate matrices as

$$L_l(\mathbf{x}) \leq \|J_f(\mathbf{x})\|_{p,q} \leq \|\mathbf{W}_l\|_{p,q} \cdots \|\mathbf{W}_1\|_{p,q}. \tag{24}$$

Furthermore $L_l(\mathbf{x})$ can be substituted by $L_l$, the local Lipschitz constant, as the upper bound (Eq. (24)) is independent of $\mathbf{x}$. Note that this is one of the main reasons why we consider the empirical Lipschitz to better reflect the true behaviour of the function as the NN is never exposed to the entire domain $\mathbb{R}^m$ but only a small subset dependant on the data distribution.

The other reason why this upper bound is a bad estimate is that the inequality in Eq (23) is tight only when the partial derivatives are aligned, implying, $\left\|\frac{\partial \mathbf{z}_\ell}{\partial \mathbf{z}_{\ell-1}} \frac{\partial \mathbf{z}_{\ell+1}}{\partial \mathbf{z}_\ell}\right\|_2 = \left\|\frac{\partial \mathbf{z}_\ell}{\partial \mathbf{z}_{\ell-1}}\right\|_2 \left\|\frac{\partial \mathbf{z}_{\ell+1}}{\partial \mathbf{z}_\ell}\right\|_2 \quad \forall l-2 \leq \ell \leq l$. This problem has been referred to as the problem of mis-alignment and is similar to quantities like layer cushion in Arora et al. (2018).

## D  EXPERIMENTAL DETAILS

**WideResNet-28-10** We use a standard WideResNet with 28 layers and a growth factor of 10. In total, the network has 36,539,124 trainable parameters. The network is the standard configuration with batchnorm and ReLU activations and is trained with a weight decay of $1e-4$. The learning rate was multiplied by 0.2 after 60, 120, and 160 epochs respectively.

**ResNet-110** The ResNet-110 is a standard 110 layered ResNet with batch Norm and ReLU and has $1,973,236$ parameters. The network is trained with SGD, an initial learning rate of 0.1, which is multiplied 0.1 after 150 and 250 epochs respectively, a weight decay of $5e-4$ and a momentum of 0.9.

**Densenet-100** The DenseNet-100 is a standard 100-layered densenet with Batchnorm and ReLU and has a total of $800,032$ trainable parameters. The network is trained with SGD, an initial learning rate of 0.1, which is multiplied by 0.1 after 150 and 250 epochs respectively, a weight decay of $1e-4$, and a momentum of 0.9.

**VGG19** The VGG19 model is the standard 19-layered VGG model with Batchnorm and ReLU. It has a total of $20,548,392$ trainable parameters and is trained with SGD with a momentum of 0.9 and a weight decay of $5e-4$. The initial learning rate is 0.1 and is multiplied by 0.1 after 150 and 250 epochs respectively. For the shattering experiments, we used the same architecture and the same training recipe except the initial learning rate, which was deceased to 0.01 as the model failed to learn the random labels with a large learning rate.

**AlexNet** The Alexnet model is the standard ALexNet model with $4,965,092$ trainable parameters. It was trained with SGD, with a momentum of 0.9, with an initial learning rate is 0.01, which is multiplied by 0.1 after 150 and 250 epochs respectively. The optimizer was further augmented with a weight decay rate of $5e-4$. Please refer to the next section for results on different learning rates and with and without weight decay.

### D.1  ADDITIONAL EXPERIMENTS ON GENERALIZATION

**Complexity measures**   In this section, we provide more details about the various complexity measures we used in Figure 4.

- **Spec-Fro:** $\frac{\prod_{i=1}^{L} \|\mathbf{W}_i\|_2^2 \sum_{i=1}^{L} \mathrm{srank}(\mathbf{W}_i)}{\gamma^2}$ (Neyshabur et al., 2018). This bound is the main motivation of this paper; the two quantities used to normalize the margin ($\gamma$) is the product of spectral norm i.e. $\prod_{i=1}^{L} \|\mathbf{W}_i\|_2^2$ (or worst case lipschitzness) and sum of stable rank *i.e.*, $\sum_{i=1}^{L} \mathrm{srank}(\mathbf{W}_i)$ (or an approximate parameter count like rank of a matrix).

---

[12]implying, $\max_{\mathbf{z}} \left\|\frac{\partial \phi(\mathbf{z})}{\partial \mathbf{z}}\right\|_p = 1$.

- **Spec-L1:** $\dfrac{\prod_{i=1}^{L} \|\mathbf{W}_i\|_2^2 \left(\sum_{i=1}^{L} \frac{\|\mathbf{W}_i\|_{2,1}^{2/3}}{\|\mathbf{W}_i\|_2^{2/3}}\right)^3}{\gamma^2}$, where $\|.\|_{2,1}$ is the matrix 2-1 norm. As showed by Bartlett et al. (2017), Spec-L1 is the spectrally normalized margin, and unlike just the margin, is a good indicator of the generalization properties of a network.

- **Jac-Norm:** $\sum_{i=1}^{L} \dfrac{\|\mathbf{h}_i\|_2 \|\mathbf{J}_i\|_2}{\gamma}$ (Wei & Ma), where $\mathbf{h}_i$ is the $i^{th}$ hidden layer and $\mathbf{J}_i = \frac{\partial \gamma}{\partial h_i}$ *i.e.*, the Jacobian of the margin with respect to the $i^{th}$ hidden layer (thus, a vector). Note, Jac-Norm depends on the norm of the Jacobian (local empirical Lipschitz) and norm of the hidden layers - additional data-dependent terms compared to Spec-Fro and Spec-L1, thus captures a more realistic (and optimistic) generalization behaviour.

For better clarity regarding Figure 4, we provide the 90 percentile for each of these histograms in Table 5. As the plots and the table show, both SRN and SN produces a much smaller quantity than a Vanilla network and in 7 out of the 9 cases, SRN is better than SN. The difference between SRN and SN is much more significant in the case of Jac-Norm. As this depend on the empirical lipschitzness, it provides the empirical validation of our arguments in Section 3.

| Model | Algorithm | Jac-Norm | Spec-$L_1$ | Spec-Fro |
|---|---|---|---|---|
| | Vanilla | 17.7 | $\infty$ | $\infty$ |
| ResNet-110 | Spectral (SN) | 17.8 | 10.8 | 7.4 |
| | SRN-30 | **17.2** | **10.7** | **7.2** |
| | Vanilla | 16.2 | 14.60 | 11.18 |
| WideResNet-28-10 | Spectral (SN) | 16.13 | 7.23 | 4.5 |
| | SRN-50 | 15.8 | 7.3 | 4.5 |
| | SRN-30 | **15.7** | **7.20** | **4.4** |
| | Vanilla | 19.2 | $\infty$ | $\infty$ |
| Densenet-100 | Spectral (SN) | 17.8 | 12.2 | 9.4 |
| | SRN-50 | 17.6 | 12 | 9.2 |
| | SRN-30 | **17.7** | **11.8** | **9.0** |

Table 5: Values of 90 percentile of $\log$ complexity measures from Figure 4. Here $\infty$ refers to the situations where the product of spectral norm blows up. This is the case in deep networks like ResNet-110 and Densenet-100 where the absence of spectral normalization (Vanilla) allows the product of spectral norm to grow arbitrarily large with increasing number of layers. Lower is better.

**Alexnet experiments:** Figure 6 shows the test error and generalization error of Alexnet trained with a large learning rate of $0.1$. Note that, the model fails to learn completely without weight decay. Generalisation Error decreases monotonically with decreasing $c$ in the stable rank constraint. Test error is the lowest for $c = 0.5$. The constraint becomes too aggressive for even $c$ lower than that. The slightly more interesting observation is that having a weight decay actually hurts generalization error while it has a slightly positive effect on test error.

**Low Learning Rate** Here, we train a WideResnet-28-10 with SRN, SN, and vanilla methods with an $lr = 0.01$ and weight decay of $5 \times 10^{-4}$ on randomly labelled CIFAR100. for 50 epochs. The results are shown in Table 6 and it further supports that SRN is more robust to random noise than SN or vanilla methods.

| Stable-30 | Spectral | Vanilla |
|---|---|---|
| 29.04 | 17.24 | 1.22 |

Table 6: Training Error for WideResNet-28-10 on CIFAR100 with randomized labels, low lr= 0.01, and with weight decay. (Higher is better.)

**With and without weight decay** In Figure 7a, we show the training error of Alexnet trained with SGD with and without weight decay ($= 5e - 4$) with a learning rate of $0.01$. Again, we see that a

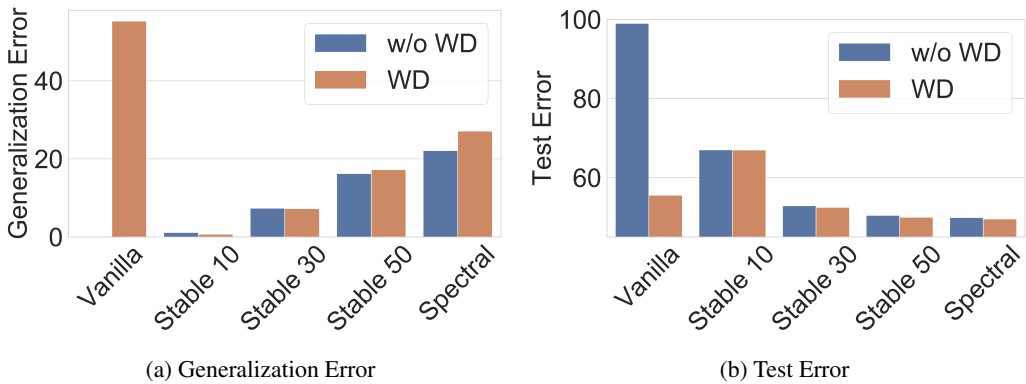

(a) Generalization Error                    (b) Test Error

Figure 6: Test Error and Generalization Error of AlexNet trained with SGD with $lr = 0.1$ on (clean) CIFAR-100. (Lower is better

more aggressive stable rank constraint decreases fitting the random data . Similar results are seen for ResNet-110 in Figure 7b.

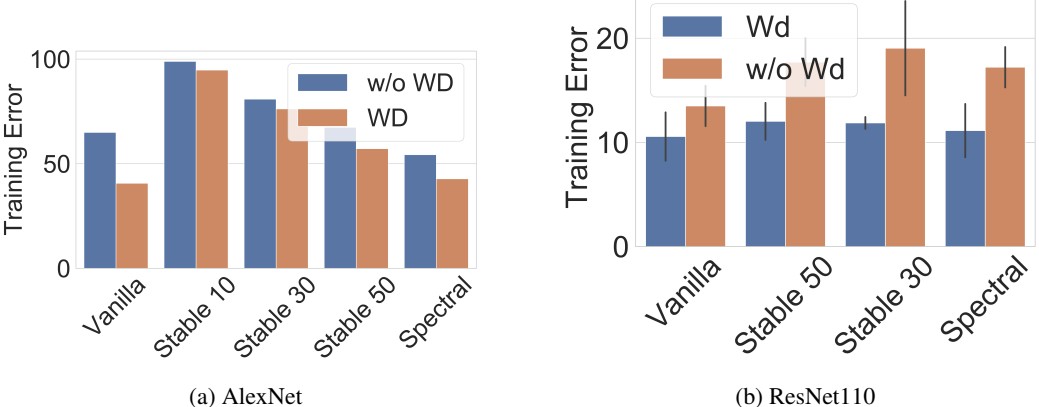

(a) AlexNet                    (b) ResNet110

Figure 7: Training error on randomly labelled CIFAR-100 with a learning rate of 0.01 and with/ with out weight decay. (Higher is better.

**Low Learning Rate, with and without weight decay on clean CIFAR100**   In Appendix D.1, we show the test accuracies for the clean data with the same confifuration as in Table 1. This corresponds to the hihgly non-generelizable learning setting.

|  | Vanilla | Spectral | Stable-50 | Stable-30 |
|---|---|---|---|---|
| W/o WD | $69.2 \pm 0.5$ | $69 \pm 0.1$ | $69.1 \pm 0.85$ | $69.3 \pm 0.4$ |
| With WD | $70.4 \pm 0.3$ | $71.35 \pm 0.25$ | $70.6 \pm 0.1$ | $70.6 \pm 0.1$ |

Table 7: Clean Test Accuracy on CIFAR10. The learning configuration corresponds to the non-generelizable settings with high learning rate. The corresponding shattering experiments for this setting are shown in Table 1.

**Training Accuracy as Stopping Criterion**   In this section we show that our regularizor performs consistently for a different stopping criterion. In particular, we use the train accuracy as a stopping criterion. For Resnet110, WideResnet-28,Densenet-100, and VGG-19 we use a train accuracy of $99\%$ as a stopping criterion and report the test accuracy when that train accuracy was achieved for the first time. For Alexnet, as SRN-30 never achieves a train accuracy higher than $55\%$, we use $55\%$

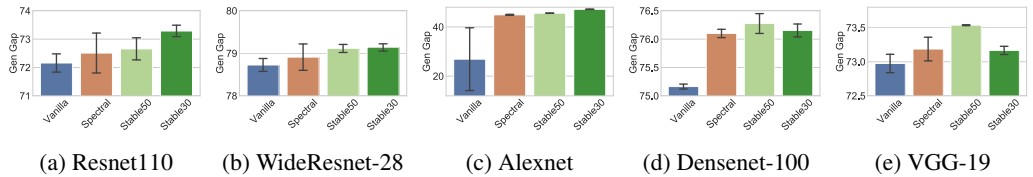

(a) Resnet110     (b) WideResnet-28     (c) Alexnet     (d) Densenet-100     (e) VGG-19

Figure 8: Test accuracies on CIFAR100 for clean data using a stopping criterion based on train accuracy. Higher is better.

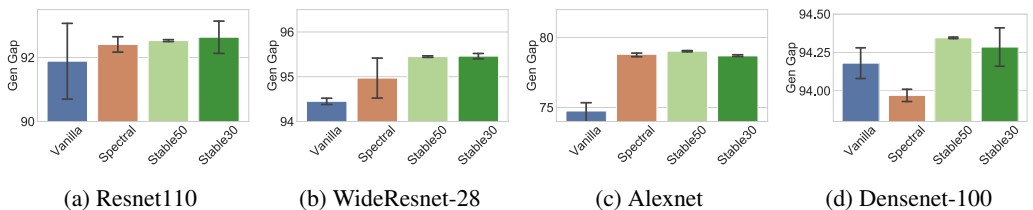

(a) Resnet110     (b) WideResnet-28     (c) Alexnet     (d) Densenet-100

Figure 9: Test accuracies on CIFAR10 for clean data using a stopping criterion based on train accuracy. Higher is better.

as the stopping criterion and plot the test accuracies in Figure 8. Our results show that SRN-30 and SRN-50 outperform SN and vanilla consistently. In Figure 9, we show similar plots for CIFAR10.

**CIFAR10 experiments** In this section, we plot results on CIFAR10 trained using ResNet-110, Desnenet100, WideResNet-28, and Alexnet. In Figure 9, we plot the test accuracy on clean CIFAR-10 with the training accuracy as the stopping criterion. For all models other than Alexnet, we use $99\%$ training accuracy as the criterion and for Alexnet we use $85\%$. In Figure 10, we plot the test accuracy on clean CIFAR10 using the number of epochs as the stopping criterion. The results here are consistent with those in the main paper in that SRN outperforms the vanilla and SN.

In Figure 11, we plot the training accuracy on CIFAR10 when the labels are randomized for Resnet100, and Alexnet. SRN-50 and SRN-30 are much better than Vanilla and SN in this case.

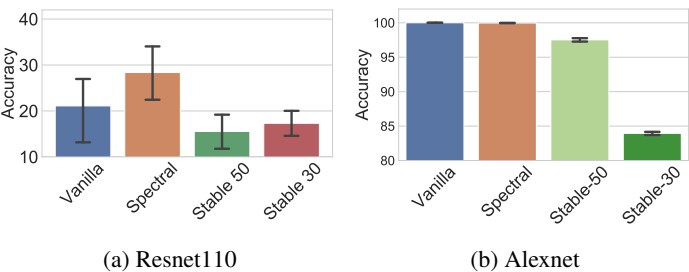

(a) Resnet110        (b) Alexnet

Figure 11: Training accuracy on randomly labelled CIFAR-10 (Lower is better).

# E  ADDITIONAL EXPERIMENTS ON GANS

## E.1  GAN EXPERIMENTAL SETUP

**Datasets and Network Architectures** Each of the CIFAR datasets contain a total of $50,000$ RGB images in the training set, where each image is of size $32 \times 32$, and a further $10,000$ RGB images of the same dimension in the test set. The CelebA dataset contains more than 200K images scaled to a size of $64 \times 64$. The model architecture for both the generator and the discriminator was chosen to be a 32 layered ResNet (He et al., 2016) due to its previous superior performance in other works (Miyato et al., 2018). We use Adam optimizer (Kingma & Ba, 2014) which depends on three main hyper-parameters $\alpha$- the initial learning rate, $\beta_1$- the first order moment decay rate and $\beta_2$- the second

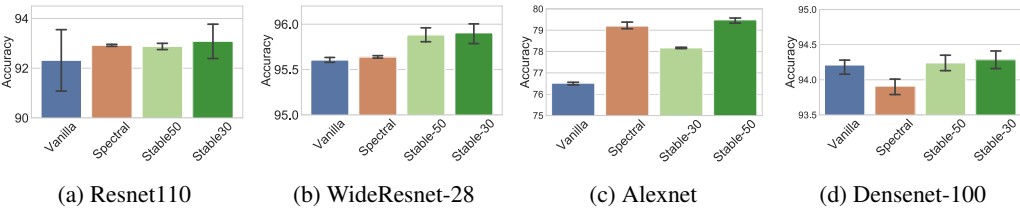

| (a) Resnet110 | (b) WideResnet-28 | (c) Alexnet | (d) Densenet-100 |

Figure 10: Test accuracies on CIFAR10 for clean data using the number of epochs as a stopping criterion. Higher is better.

order moment decay rate. We cross-validate these parameters in the set $\alpha \in \{0.0002, 0.0005\}$, $\beta_1 \in \{0, 0.5\}$, $\beta_2 \in \{0.9, 0.999\}$ and chose $\alpha = 0.0002$, $\beta_1 = 0.0$ and $\beta_2 = 0.999$ which performed consistently well in all of the experiments.

**GAN objective functions**   In the case of conditional GANs (Mirza & Osindero, 2014), we used the conditional batch normalization (Dumoulin et al., 2017) to condition the generator and the projection discriminator (Miyato & Koyama, 2018) to condition the discriminator. The dimension of the latent variable for the generator was set to 128 and was sampled from a zero mean and unit variance Gaussian distribution. For training the model, we used the hinge loss version of the adversarial loss (Lim & Ye, 2017; Tran et al., 2017) in all experiments except the experiments with WGAN-GP. The hinge loss version was chosen as it has been shown to give consistently better performance in previous works (Zhang et al., 2018; Miyato et al., 2018). For training the WGAN-GP model, we used the original loss function as described in Gulrajani et al. (2017).

**Evaluation Metrics**   We use *Inception* (Salimans et al., 2016) and *Frechet Inception Distance* (FID) (Heusel et al., 2017) scores for the evaluation of the generated samples. For measuring the inception score, we generate $50,000$ samples, as was recommended in Salimans et al. (2016). For measuring FID, we use the same setting as Miyato et al. (2018) where we sample $10,000$ data points from the training set and compare its statistics with that of $5,000$ generated samples. In addition, we use a recent evaluation metric called Neural divergence score Gulrajani et al. (2019) which is more robust to memorization. The exact set-up for the same is discussed below. In the case of conditional image generation, we also measure Intra-FID (Miyato et al., 2018), which is the mean of the FID of the generator, when it is conditioned over different classes. Let $\mathrm{FID}(\mathcal{G}, c)$ be the FID of the generator $\mathcal{G}$ when it is conditioned on the class $c \in \mathcal{C}$ (where $\mathcal{C}$ is the set of classes), then, $\mathrm{Intra}\, \mathrm{FID}(\mathcal{G}) = \frac{1}{|\mathcal{C}|} \mathrm{FID}(\mathcal{G}, c)$

**Neural Divergence Setup**   We train a new classifier inline with the architecture in Gulrajani et al. (2019). It includes three convolution layers with 16, 32 and 64 channels, a kernel size of $5 \times 5$ and a stride of 2. Each of these layers are followed by a Swish activation (Ramachandran et al., 2018) and then finally a linear layer that gives a single output. The network is initialized using normal distribution with zero mean and the standard deviation of 0.02, and trained using Adam optimizer with $\alpha = 0.0002$, $\beta_1 = 0.$, $\beta_2 = 0.9$ for a total of $100,000$ iterations with minibatch of 128 generated samples and 128 samples from the test set[13]. We use the standard WGAN-GP loss function, $\log\left(1 + \exp\left(f\left(\mathbf{x}_{\mathrm{fake}}\right)\right)\right) + \log\left(1 + \exp\left(-\mathbf{x}_{\mathrm{real}}\right)\right)$, where $f$ represents the network described above. Finally, we generate 1 Million samples from the generator and report the average $\log\left(1 + \exp\left(f\left(\mathbf{x}_{\mathrm{fake}}\right)\right)\right)$ over these samples. Higher average value implies better generation as the network in this case is unable to distinguish the generated and the real samples.

### E.2   More Empirical Lipschitz plots

For the purpose of analysis, Figure 13b and 14b shows eLhist for pairs where each sample either comes from the true data or from the generator, and we observe a similar trend. To verify that same results hold in the conditional setup, we show comparisons for GANs with projection discriminator (Miyato & Koyama, 2018) in Figure 12, 13a and 14a, and observe a similar trend. Further, to see the value of the local Lipschitzness in the vicinity of real and generated samples we also plot the norm of the

---

[13]For CelebA, we used the training set.

Jacobian in Figure 15 and 16 in Appendix E.2 and observe mostly a similar trend. In Appendix E.3 (Figure 17), we also show that the discriminator training of SRN-GAN is more stable than SN-GAN.

**Conditional GANs** Figure 12 shows the eLihst of conditional GANs with projection discriminator (Miyato & Koyama, 2018).

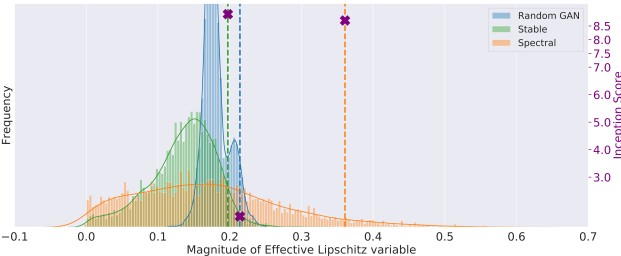

Figure 12: Comparison: **eLhist** of the discriminator in the conditional GAN setting with projection discriminator on CIFAR100.

**Empirical Lipschitzness between real samples and between fake samples.** Figure 13 shows the histogram of eLhist of the discriminator for pairs of fake samples i.e. samples generated by the generator. Figure 14 shows eLhist of the discriminator when samples came from the dataset.

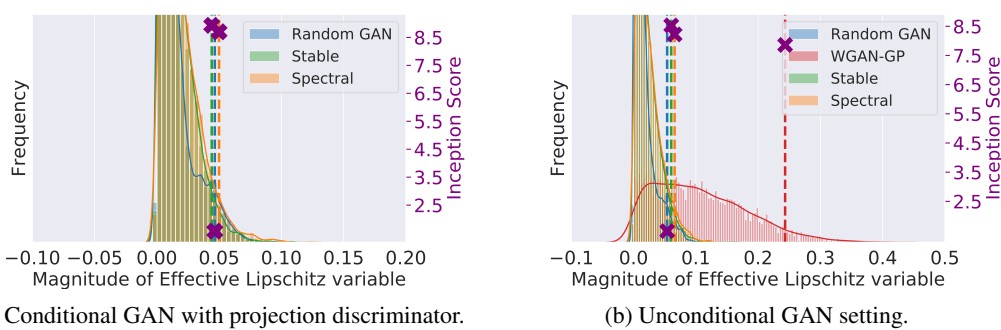

(a) Conditional GAN with projection discriminator.  (b) Unconditional GAN setting.

Figure 13: Comparison: **eLhist** of the discriminator for pairs of samples selected from the generator on CIFAR10

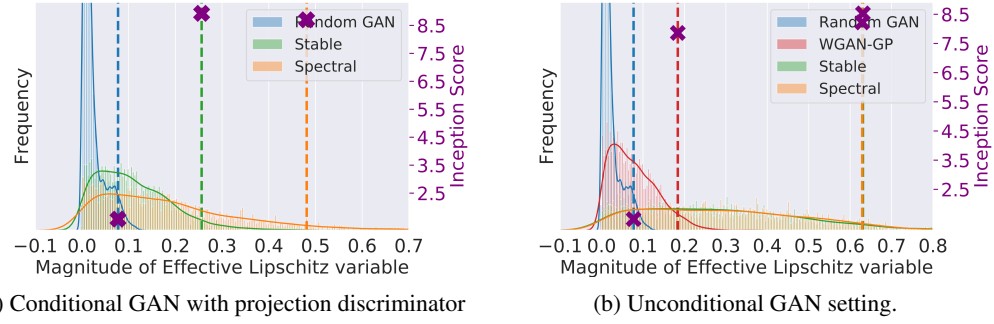

(a) Conditional GAN with projection discriminator  (b) Unconditional GAN setting.

Figure 14: Comparison: **eLhist** of the discriminator for pairs of samples from the real distribution on CIFAR10.

**Jacobian norm in the vicinity of the points** Here we compare the Jacobian of the discriminator of the trained models in the vicinity of the samples from the generator and the real dataset. This is a penalized measure in various algorithms Gulrajani et al. (2017); Petzka et al. (2018) (often referred to as local perturbations) and was independently proposed by Kodali et al. (2018). Figure 15

and Figure 16 show the histogram of the norm of the Jacobian of the discriminator in the vicinity of the generated and the real samples, respectively. To generate these plots, $2,000$ samples were used from the respective distributions. It is interesting to note that the norm is the same for the points in the vicinity of the real data points and the generated data points for the Stable Rank Normalization GAN (SRN-GAN) as well for WGAN-GP whereas it varies between fake and real samples for Spectral Normalization GAN (SN-GAN).

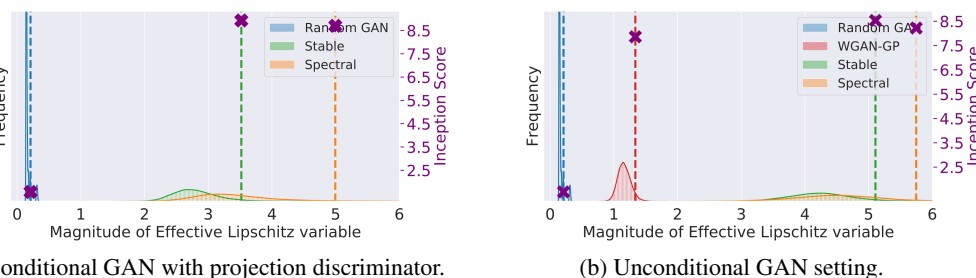

(a) Conditional GAN with projection discriminator.          (b) Unconditional GAN setting.

Figure 15: Jacobian norm of the discriminator in the neighbourhood of the samples from the *generator* trained on CIFAR10.

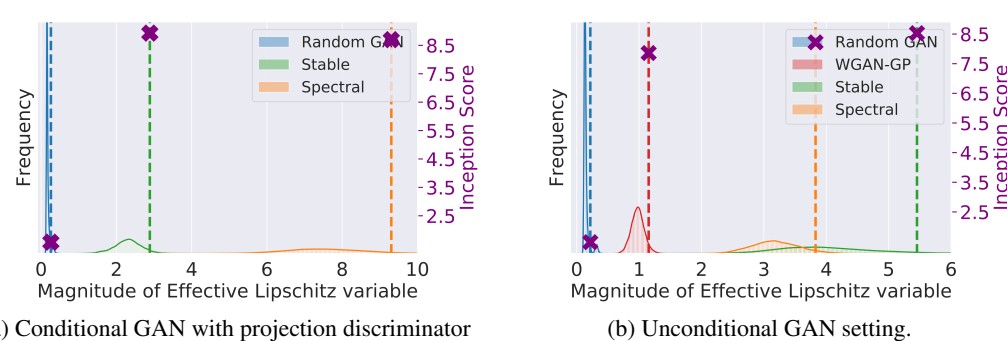

(a) Conditional GAN with projection discriminator          (b) Unconditional GAN setting.

Figure 16: Jacobian norm of the discriminator in the neighbourhood of the samples from the *real dataset* (CIFAR10).

### E.3 TRAINING STABILITY

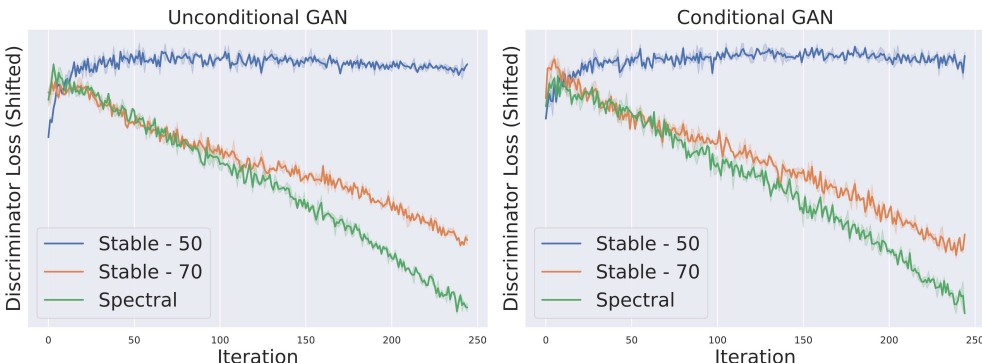

Figure 17: Loss incurred by the discriminator. The loss of SRN-GAN with the stable rank constraint of 70 is shifted upwards by 0.2 so that we can compare the change of the loss during training as opposed to the absolute magnitude of the loss.

**Training Stability**    In Figure 17 we show the discriminator loss during the course of the training as an indicator of whether the generator gets sufficient gradient during training or not. These plots clearly suggest that the discriminator loss is more consistent for SRN than the SN.

# F    EXAMPLES OF GENERATED IMAGES

## F.1    CELEBA IMAGES

For these images, we generated 100 images from the respective models and hand-picked the 10 best images in terms of visual quality.

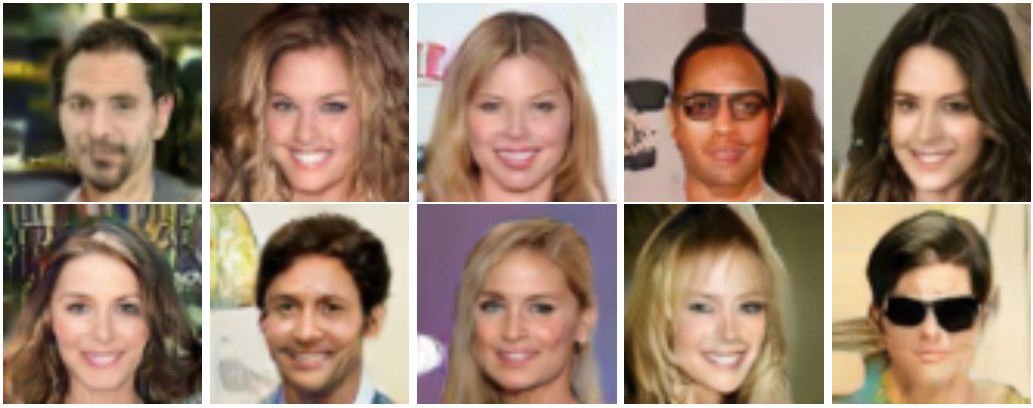

Figure 18: Image samples generated from the unconditional SRN-GAN.

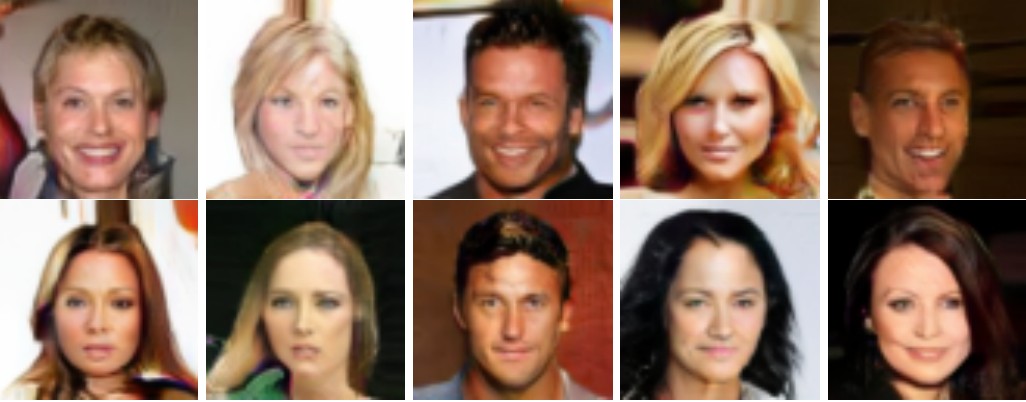

Figure 19: Image samples generated from the unconditional SN-GAN.

## F.2 CIFAR10-UNCONDITIONAL GAN

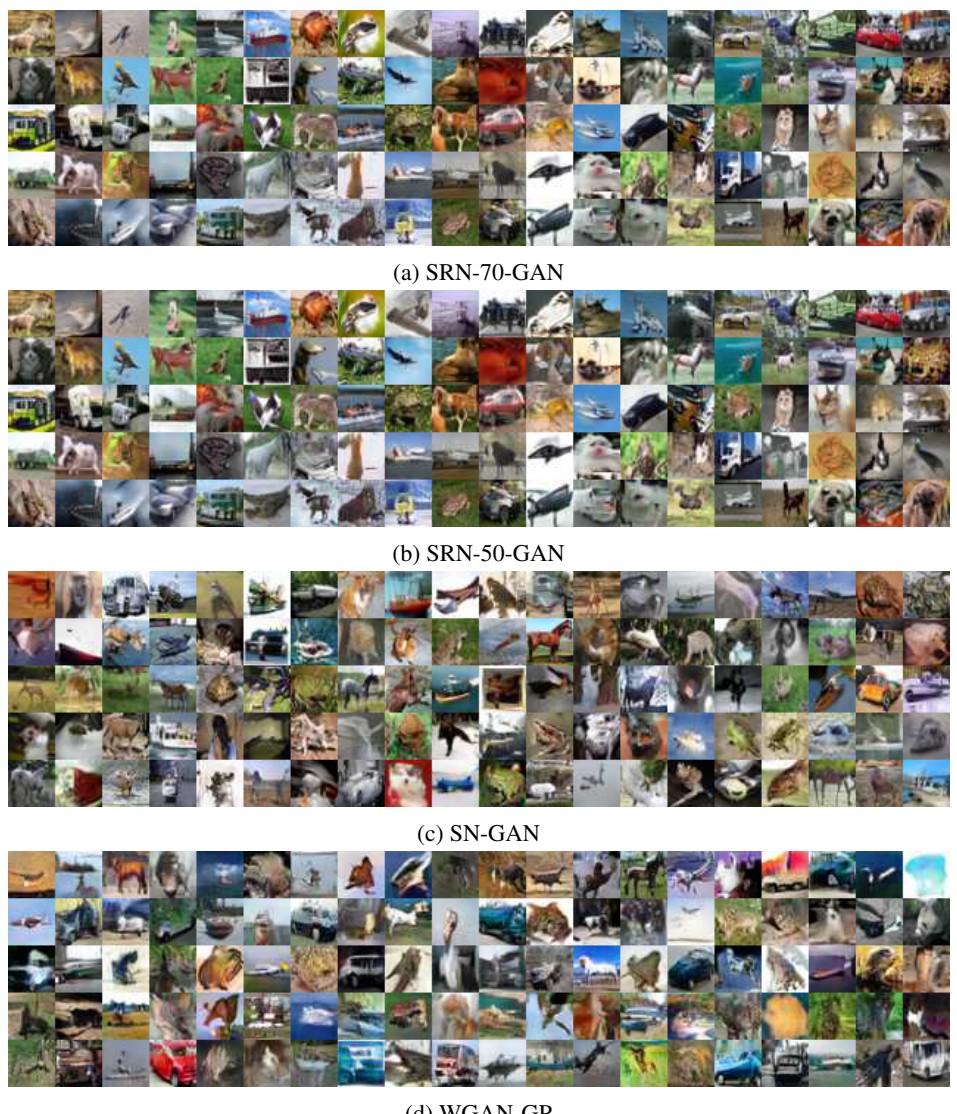

(a) SRN-70-GAN

(b) SRN-50-GAN

(c) SN-GAN

(d) WGAN-GP

Figure 20: Image samples generated from the unconditional SRN-GAN, SN-GAN, and WGAN-GP.

### F.3 CIFAR10-CONDITIONAL SRN-GAN

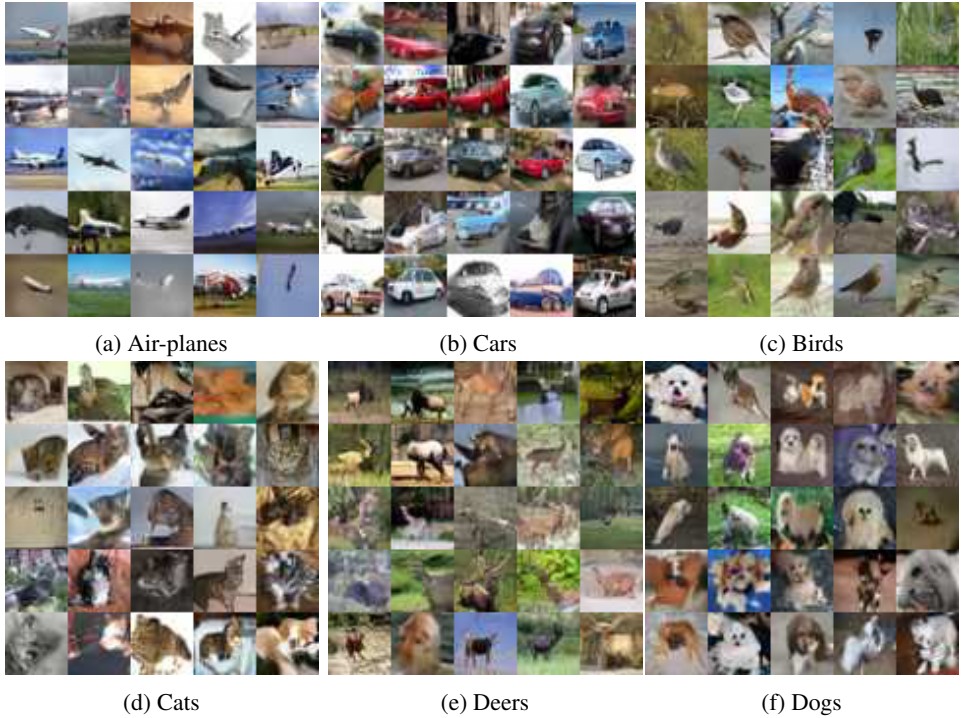

|   |   |   |
|:-:|:-:|:-:|
| (a) Air-planes | (b) Cars | (c) Birds |
| (d) Cats | (e) Deers | (f) Dogs |

Figure 21: Image samples generated from the conditional SRN-GAN with projection discriminator.

