# OpenReview forum: "Stable Rank Normalization for Improved Generalization in Neural Networks and GANs"
_ICLR.cc/2020/Conference — Accept (Spotlight)_

### Official Review · AnonReviewer2 · 2019-10-18
**Official Blind Review #2**

**Rating:** 8

**Review:**

While spectral normalization is often used to improve generalization by
directly bounding the Lipschitz constant of linear layers, recent works have
highlighted alternate methods that aim to reduce generalization error.  This
paper shows how to implement these "stable rank" normalizations with little
computational overhead.  The authors then apply the method to a wide variety of
classification and GAN problems to show the benefits of stable rank
normalization.

This is a good paper and can be accepted.  The added value comes from their
Thm. 1, where they detail precisely how to project a real matrix onto one of
lower srank while preserving the largest k eigenvalues.  The spectral preservation
k seems to be a new feature of their method. Full proofs and additional results are
provided in appendices.  There seems to be enough information to implement
the described methods.

The paper is carefully written and introductory sections do a great job of putting
the problem in perspective.  Very few typos ("calssification", run a spell check).

--------- Fun to think about ------ here are some extra comments

Some related older introductory approaches could also be quickly mentioned:
 - linear layers represented as "bottlenecks" to enforce low rank explicitly
 - or solving in manifold of reduced-rank matrices directly

For simplicity, they target the same srank r=c*min(m,n) for all layers, even though only the sum
of sranks is important.  For CNNs with only a few linear layers is there any observable
difference by lightly deviating from this?  Does the first linear layer typically contribute
the lion's share to the sum of sranks?

It is interesting that by only addressing the linear layers of deep CNNs they
are able to see consistent improvements. [i.e. 3 linear layers after 101 CNN layers].
 This makes me wonder whether future work will also address how "stable rank"
concepts might be extended to the convolutional layers.  As a starting point, spectral
values of the block-circulant matrices corresponding to convolutions have been
described [ Sedghi et al. "Singular Values of Convolutional Layers" ].


**Experience Assessment:**

I have read many papers in this area.

**Review Assessment: Checking Correctness Of Derivations And Theory:**

I assessed the sensibility of the derivations and theory.

**Review Assessment: Checking Correctness Of Experiments:**

I assessed the sensibility of the experiments.

**Review Assessment: Thoroughness In Paper Reading:**

I read the paper at least twice and used my best judgement in assessing the paper.

---

> ### Author Response · Authors · 2019-11-13
> **Response to Reviewer #2**
>
> First of all, we would like to thank the reviewer for reading our paper very carefully, appreciating our technical contribution, providing extremely positive comments, and  commenting on the importance of Theorem 1. We also appreciate that the reviewer finds that the introduction did a great job and also that our paper contains enough information to implement the approach. All these comments are extremely encouraging.
>
> Below we provide answers to some of  the comments given by the reviewer.
>
> >> Some related older introductory approaches could also be quickly mentioned:
>  - linear layers represented as "bottlenecks" to enforce low rank explicitly
>  - or solving in manifold of reduced-rank matrices directly
>
> —  Thank you for the suggestion. We will definitely add a bit of surrounding literature on the following subjects: a) low rank weights and its application (mainly to compression), low rank activations, and optimization on the manifold of low rank (and PSD) matrices.
>
> >>  For simplicity, they target the same srank r=c*min(m,n) for all layers, even though only the sum of sranks is important.  For CNNs with only a few linear layers is there any observable difference by lightly deviating from this?
>
> —   We would like to clarify that when we say linear layers, we also refer to the linear transformations in all layers (i.e. those present in the convolutional layer as well) . So, our method is indeed applied to all the layers (both fully connected and convolutional) of the network. We computed m and n here the same as computed in Miyato et. al.  We did not vary the stable rank constraint for each layer, we keep it the same to avoid too many combinations but it could be the objective of a future study.
>
> >>  ..whether future work will also address how "stable rank" concepts might be extended to the convolutional layers.  As a starting point, spectral values of the block-circulant matrices corresponding to convolutions have been described [ Sedghi et al. "Singular Values of Convolutional Layers" ].
>
> —  As mentioned earlier, we did use SRN for the convolutional layer as well. We will modify the text to make it explicit in order to avoid any confusion. Our singular value extraction is similar to the one presented in the spectral norm paper. Certainly, future work would consist of using the algorithm in Sedghi et. al. to properly extract the singular values of Convolutional layers and using different k’s (partitioning index) to run the experiments.
>
> Miyato, Takeru, et al. "Spectral normalization for generative adversarial networks." arXiv preprint arXiv:1802.05957 (2018).

---

> > ### Comment · AnonReviewer2 · 2019-11-13
> > **convolution heuristic (my bad).**
> >
> > I guess I should have known that references to Miyato's paper implied that you were using an empirical heuristic to treat convolutional layers as linear transforms. Instead I wrongly guessed that the block-sparse linear matrix corresponding to a convolution would add another projection step to Thm 1, so you did not do convolutional layers.  My naive expectation is that accuracy and even speed won't change much moving from heuristic to exact spectral values, but it would be nice to confirm at some point.

---

### Official Review · AnonReviewer1 · 2019-10-21
**Official Blind Review #1**

**Rating:** 8

**Review:**

Stable Rank Normalization for Improved Generalization in Neural Networks and GANs

Summary:

This paper proposes to normalize network weights using the Stable Rank, extending the method of spectral normalization. The stable rank is the ratio of the squared frobenius norm to the squared spectral norm (the top singular value). The authors motivate this decision in the context of lipschitz constraints and noise sensitivity. The proposed method (combined with spectral norm as SRN alone does not explicitly subsume SN) is tested for both classification (using a wide range of popular models on CIFAR) and GAN training. Performance (classification accuracy and FID/IS) is measured and several auxiliary investigations are performed into generalization bounds, sample complexity, and sensitivity to hyperparameters.

My Take:

This paper motivates and presents an interesting extension of spectral norm, and evaluates it quite well with thorough experiments in a range of settings. The method looks to be reasonably accessible to implement, although its compute cost is not properly characterized and some details (like the orthogonalization step necessary in power-iteration for more than one SV) seem to be omitted. My two main concerns are that the results, while good, are not especially strong (the relative improvement is not very high) and that the paper could be made substantially more concise to fit within the 8 page soft limit (I felt there was plenty of material that could be moved to the appendix). All in all this is a reasonably clear accept to me (7/10) that with some cleanup could be a more solid 8, and I argue in favor of acceptance.

Notes

-The paper should characterize the runtime difference between SRN and SN. It is presently unclear how computationally intensive the method is. What is the difference in time per training iteration? The authors should also indicate their hardware setup and overall training time.

-I found table 1 confusing as it lacks the test error. Are the test errors the same for all these models and the authors are just showing that for certain settings the SRN models have higher training error? If there is a difference in testing error, then this table is misleading, as one cares little about the training error if the test errors vary. If the test errors are approximately the same, then why should I care if the training error is higher? This would just be a way to decrease the stated “generalization gap,” which is not necessarily indicative of a better model (vis-à-vis the commonly held misconception that comparing training error between models is properly indicative of relative overfitting).

-Nowhere (that I could spot) in the body of the paper is it explained what “Stable-50”, “SRN-50”, and “SRN-50%” are. I assume these all mean the same thing and it refers to the choice of the c hyperparameter, but this should be explicitly stated so that the reader knows which model corresponds to which settings.

Minor

-The footnotes appear to be out of order,  footnote 1 appears on page 9

-There are typos such as “simiplicity,” please proofread thoroughly.

**Experience Assessment:**

I have published in this field for several years.

**Review Assessment: Checking Correctness Of Derivations And Theory:**

I assessed the sensibility of the derivations and theory.

**Review Assessment: Checking Correctness Of Experiments:**

I carefully checked the experiments.

**Review Assessment: Thoroughness In Paper Reading:**

I read the paper thoroughly.

---

> ### Author Response · Authors · 2019-11-13
> **Response to Reviewer #1**
>
> We would like to thank the reviewer for taking time to read the paper carefully and for the encouraging comments. We have fixed the typos in the updated version. We hope our comments below addresses your questions.
>
> >> The method looks to be reasonably accessible to implement, although its compute cost is not properly characterized and some details (like the orthogonalization step necessary in power-iteration for more than one SV) seem to be omitted.
>
> — The method is indeed very simple to implement. It requires a mere addition of a few lines to the existing code of the widely used spectral normalization (SN) algorithm. The exact computational complexity of doing SRN i.e. to obtain $W_f$ in Line 10 in Algorithm 2 is $\mathcal{O}(mn)$ where m,n are the dimensions of the matrix. The SRN and the SN algorithm only requires one singular value. However, to use the other variants  (using the partitioning index k) that can be formed from Theorem 1, the cost would scale as $\mathcal{O}(mnk)$ where k is the number of singular values we wish to preserve.
>
> >>  What is the difference in time per training iteration? The authors should also indicate their hardware setup and overall training time.
>
> — Running the experiment on  a NVIDIA V100 for a ResNet110, amount of time taken for one epoch with a batch size of 128 is as follows - a. Vanilla - 187.9 sec (15.5 hrs in total), b. Spectral - 207.9 sec (17 hrs in total), c. Stable-50 - 227.8 s (19 hrs in total), d. Stable-30 - 234 s (19.5 hrs in total). In brackets is the time taken to run an a whole experiment on clean labels.
>
> >>  Table 1 confusing as it lacks the test error.
>
> — The particular example in Table 1 is for randomized labels with low learning rate, with and without weight decay. The test error here is thus simply 0.01 which is the chance of getting the label right by guessing randomly. These experiments are in line with the shattering experiments present in Figure 3 for the harder to generalize setting (low lr). We can see  that this is the harder to generalize setting (also noted in Wei et. al. 2019) as the training errors here are indeed much lower than those in Figure 3 eg. 40% error in high lr vs 10% error in low lr for vanilla network.
>
> However, we are also $\textbf{including the test error of the same configuration on clean labels}$ below. Note that they are almost the same and they are slightly worse than the high learning rate configuration (Figure 2). This again indicates that this is the harder to generalize setting.
>
>
> ____________________________________________________________________
>                   |  Vanilla     | Spectral       | Stable-50    | Stable-30 |
> W/o WD   | 69.2 ±0.5  |69 ±0.1          | 69.1 ±0.85   | 69.3 ±0.4  |
> With WD  | 70.4 ±0.3  | 71.35 ±0.25  | 70.6 ±0.1     | 70.6 ±0.1  |
>
>
> >> Concise within 8 pages.
>
> —  We thank the reviewer for this suggestion. We had previously used all the pages available to us to make it detailed and easy to read. However, in the next version we will make it more concise, fix the spelling mistakes and remove other errors like using different names for the same model without mentioning it. While we haven’t been able to fit it into 8 pages, we have shortened it a bit by moving slightly less necessary things to the appendix. If there is something in particular you feel could be easily moved to the Appendix please do let us know.

---

### Official Review · AnonReviewer3 · 2019-10-24
**Official Blind Review #3**

**Rating:** 6

**Review:**

This paper proposes normalizing the stable rank (ratio of the Frobenius norm to the spectral norm) of weight matrices in neural networks. They propose an algorithm that provably finds the optimal solution efficiently and perform experiments to show the effectiveness of this normalization technique.

Stable rank of the weight matrix is an interesting quantity that shows up in several generalization bounds. Therefore, regularizing such measure could potentially help with generalization. Authors discuss this clearly and they provide an algorithm that provably finds the projection. I enjoyed reading this part of the paper. The only question that I have from this part is the role of partitioning index. It looks like it is not really being used in the experiments later. Is that right? What is the importance of adding it to the paper if it is not being used?

My main issue is with the empirical evaluation of the normalization technique. I am not an expert in GANs so I leave that to other reviewers to judge. Experiments on random labels and looking at different generalization measures are all nice but they are not sufficient for showing that this normalization technique is actually useful in practice. Therefore, I suggest authors to put more emphasize at showing how their regularization can improve generalization in practice. My suggestions:

- Authors only provided experiments on CIFAR100 dataset to support their claim on improving generalization. I suggest adding at least one other dataset (CIFAR10, or even better imagenet) to improve their empirical results.

- Unfortunately, there are two major issues with the current CIFAR100 results: 1) the accuracies reported for ResNet and DenseNet are too low compare what is reported in the literature. Please resolve this issue. 2) The current result is with training with a fixed number of epochs. Instead, train with a stopping criterion based on the cross-entropy loss on the training set and use the same stopping criterion for all models. Also, add the plots that show training and test errors based on the #epochs.


Overall, I think the paper is interesting but the empirical results are not sufficient to support the main claim of the paper (improving generalization). I'm willing to increase my score if authors apply the above suggestions.


*************************

After author rebuttals:

Authors have address my concerns adequately in the last revision and improved the experiment section. Therefore, I increase my score to 6.

**Experience Assessment:**

I have published in this field for several years.

**Review Assessment: Checking Correctness Of Derivations And Theory:**

I assessed the sensibility of the derivations and theory.

**Review Assessment: Checking Correctness Of Experiments:**

I carefully checked the experiments.

**Review Assessment: Thoroughness In Paper Reading:**

I read the paper thoroughly.

---

> ### Author Response · Authors · 2019-11-13
> **Additional Experiments and response to Reviewer #3**
>
> Thank you very much for reading the paper in detail , finding it interesting, and providing questions and suggestions. Below we answer all your questions and hopefully it will convince you to change your score. To summarize, there are few concerns that you raised: (1)  stopping criterion based on cross entropy; (2) empirical evaluation; (3) accuracy of the models compared to sota in the literature; (4) new dataset; and (5). role of the partitioning index
>
>
> >> Stopping criterion:  We are not sure how using the training loss as a stopping criterion would be useful or insightful. Nonetheless we think this poses two problems.
>
>                 i) Different models trained for the same experiment, es-specially the ones on  randomized label, takes a long time for $\textbf{all of them}$ to reach the same training accuracy/loss. From our experiments, some instances will never reach the training accuracy of 99% (and hence we will never know what is the right time to stop) whereas some of them will reach it within 500 epochs. So, it doesn’t give us a stopping criterion which can be executed efficiently. We believed a fairer stopping criterion is to stop them after a reasonably large number of epochs.
>
>                 ii) As we are working with the generelization error of a class of models, we assume our model class to be the set of ResNets (or WRNs, Densenets etc) with the given architecture trained for N epochs. This is a $\textbf{valid hypothesis class and is used commonly in practice}$. It is very uncommon to use a stopping criterion that depends on the training data and in this case, using one that depends on the validation data will not work as the validation loss on random data will stay constant.
>
> >> New dataset: Thank you for the suggestion. We are have reported $\textbf{ResNet-110 and Alexnet results on CIFAR10}$ now in Appendix D.1. In response to Reviewer 1’s comment, we have also performed $\textbf{more experiments on ResNet-110 with low learning rates}$, with and without weight decay on clean CIFAR-100. These results show consistent advantage in favour of SRN as a regularizor across models, datasets, and learning hyperparameters.
>
>
> >> Accuracy of the models used:    The accuray obtained for ResNet-110 on CIFAR100 for our network with $\textbf{1.9M parameters is 72.5}$ which is better than what ResNet110 obtains on https://github.com/bearpaw/pytorch-classification  ~ $\mathbf{71.14\%}$.  This is the best that can be obtained with this network configuration and size. Other ResNets that do obtain better accuracy has almost $\textbf{25 times the number of parameters~(45 M parameters)}$ as they have 4 times the number of output filters on each convolution layer, making it computationally very prohibitive. We believe it is fair comparison given that ResNet-110 with 1.9M parameters is a very standard resnet to test algorithms on. For a more wider network, one can look at the results on WideResNet.
>
>      The implementation of densenet-BC (L=100) at https://github.com/bearpaw/pytorch-classification achives an error of 22.88, densenet121~(ours is densenet100) at https://github.com/weiaicunzai/pytorch-cifar100 achieves an error of 23.99, and densenet(BC, L=100,K=12) at   https://github.com/liuzhuang13/DenseNet   achieves an error of 22.27. Our error for the vanilla model is at 24.74 which is slightly lower than theirs on densenet due to the absence of dropout in the model, which we did to isolate the effect of SRN. Also note that we report the  $\textbf{mean of several runs}$  whereas the accuracies reported here are mostly tuned to be the one that performs best on the validation set. If we reported the best accuracy on our models, the accuracy would be higher but there would be no statistical significance of the result.
>
> We would again stress that both densenet and ResNet $\textbf{perform within 1~2% of the widely reported accuracies}$ of these models and the comparison between the models are fair as they are carried out between the exact same architectures and learning configurations.
>
> Empirical evaluation: We use various ways to empirically show improved generalization. We agree that showing only based on the randomized labels might not be enough and that’s the primary reason why we use $\textbf{three other}$ criterions to show that SRN does improve generalization. These three additional criteria--(a), (b), and (c)-- capture the generalization properties of the network and we believe that a model performing best on all the four settings (including the randomized as well), while maintaining high accuracy, provides better generatlization.  Showing better generelization is quite hard and we aren’t aware of any other means to empirically show such behaviour and will be extremely thankful if reviewer could point out any other ways of empirically evaluating the same. We will try our best to incorporate that as well in our experiments.

---

> > ### Author Response · Authors · 2019-11-13
> > **Reply (part 2)**
> >
> > Role of partitioning index: Note our final algorithm does use partitioning index of $k = 1$. Our problem formulation is more general than that as it allows varying k while obtaining optimal solution. We provide this formulation with a hope that it will be useful for other fields as well where varying k is feasible or more important. However, in the case of deep learning, varying k is computationally  expensive as it will require obtaining top k singular vectors. We do agree that we could explore until k = 3, but that itself would have increased the number of experiments 3 times, and also the point that we are trying to make, which is to show that normalizing parameter dependent quantities found in recent generalization bounds-- “stable rank” and “spectral normalization”-- improves generalization, wouldn’t change.

---

> > > ### Comment · AnonReviewer3 · 2019-11-14
> > > **Thanks for your response**
> > >
> > > Thanks for your response. Unfortunately, some of my concerns about the experiments are not addressed:
> > >
> > > 1- #epoch as the stopping criterion: I am still worried about this choice. You have did not add any results for cross-entropy as the stopping criterion nor provided any evidence that your result is not sensitive to the choice of stopping criterion.
> > >
> > > 2- Experiments on CIFAR10 and CIFAR100 (clean labels): Thanks for adding experiments on CIFAR100. I was hopping that you would repeat the CIFAR10 experiments (clean cases) for CIFAR100 but only two architecture are reported for CIFAR100. Even in these two cases,  SRN30 and SRN30 are not clearly better than spectral normalization. This is also the case in the experiments with weight decay where spectral normalization ends up outperforming SRN30 and SRN50 in terms of the test error.

---

> > > > ### Author Response · Authors · 2019-11-15
> > > > **New Stopping criterion and more architectures for CIFAR10 experiments (clean cases)**
> > > >
> > > > Thank you for your response and suggestions. We have now incorporated your suggestions  in the following way.
> > > >
> > > > 1. #epoch as the stopping criterion. We understand that you would prefer to see the training loss as a stopping criterion as well to check if the method is sensitive to stopping criterion. We have now uploaded results where we used the training accuracy as the stopping criterion on both CIFAR100 (using ResNet110, WideResnet-28, Densenet100, Alexnet, and VGG19 in Figure 6) and CIFAF10(using ResNet110, WideResnet-28, Densenet100, and Alexnet in Figure 9). We used the accuracy where we fixed 99% train accuracy as the criterion and SRN performs better than SN and vanilla consistently with this new stopping criterion.
> > > >
> > > > 2. Experiments on CIFAR10(clean labels): Upon your suggestion, we have now also added WideResNet-28 and Densenet100. The experiments took a bit longer to finish which is why we couldn't add them in the last revision. SRN-50 and SRN-30 are  both better than SN and Vanilla on Densenet-100, WideResNet-28, and ResNet-110 (Figure 10). As you noted, it is only SRN-30 with Alexnet that performs suboptimally compared to SN. SRN is better in all the other cases here.
> > > >
> > > > We hope these new experiments will address your remaining concerns and you will reconsider your score.
> > > >
> > > > Thank you.

---

### Public Comment · ~Micah_Goldblum1 · 2019-11-08
**An Interesting Connection**

Hi Authors,
Thank you for your interesting paper.  I noticed that your work concerning stable rank normalization is related to our paper [1], which showed that effective rank does not correlate with test performance in some cases.  Please consider mentioning the relationship with our work in your next version.

[1] https://arxiv.org/abs/1910.00359

---

### Author Response · Authors · 2019-11-13
**Updated revision**

We would like to thank all the reviewers for their insightful comments and suggestions to improve our paper. We have now posted a revision with the following main revisions.

* New Experiments on CIFAR10 with Resnet110 and Alexnet on CIFAR10 both for clean and random data. (Figure 8)

* New Experiments on (clean) CIFAR100 with low learning rate, and with and without weight decay  using ResNet110. (Table 7)

* Shortened the paper a bit, corrected typos and other minor corrections.

---

### Author Response · Authors · 2019-11-15
**Revision #2 uploaded with suggestions from Reviewer#3 with more stopping criterion and model**

Upon suggestions from Reviewer 3, we have now made the following updates in the latest revisiion.

* Figure 6 (CIFAR100) and Figure 9 (CIFAR10) now use the training accuracy as a stopping criterion. We report the test accuracy when the training accuracy first reaches the stopping criterion.

* Figure 10 now includes WideResnet28 and Densenet100 in addition to Resnet110 and Alexnet.

---

### Public Comment · ~Thanh_Tung_Hoang1 · 2019-12-21
**The correctness of the intuition and some other questions**

Hi there,
Thank you for a nice paper on the generalization of neural net and GANs. However, I find the intuition that "lower stable rank leads to better generalization" is not well justified. In this paper [1] (also will be presented at ICLR 2020)
the author showed that spectral norms based generalization bounds negatively correlate to generalization. Minimizing the stable rank thus might not improve generalization as expected.

"Many norm-based measures not only perform poorly, but negatively correlate with
generalization specifically when the optimization procedure injects some stochasticity. In particular, the generalization bound based on the product of spectral norms
of the layers (similar to that of Bartlett et al. (2017)) has very strong negative
correlation with generalization." [1]

Another problem with your method is that the optimal stable rank (the stable rank that results in the best generalization)  cannot be computed exactly so it must be chosen empirically.

For the generalization of GANs, this paper [2] shows that the gradient of the optimal discriminator goes toward 0 as the two distributions become closer. So to improve generalization, the gradient of the discriminator should be pushed toward 0. The paper also shows that any discriminator with Lipschitz constant greater than 0 does not guarantee good generalization.

[1] "Fantastic Generalization Measures and Where to Find Them"
https://openreview.net/forum?id=SJgIPJBFvH&noteId=SJgIPJBFvH
[2] "Improving Generalization and Stability of Generative Adversarial Networks"
https://openreview.net/forum?id=ByxPYjC5KQ

---

> ### Author Response · Authors · 2020-04-27
> **Apologies for late reply and answer to your questions**
>
> Apologies for our late reply and thank you very much for your questions. They are all valid, and we are happy to answer them. Please do let us know if something isn’t clear.
>
> Q1: In this paper [1] (also will be presented at ICLR 2020) the author showed that spectral norms based generalization bounds negatively correlate to generalization. Minimizing the stable rank thus might not improve generalization as expected…
>
> Ans: Great question. We would like to point out that even though [1] is a great empirical analyses paper, there are a few caveats (mentioned in [1] as well) among which the most relevant to your question is that --  [1] focus only on finding the correlation between various complexity measures and the generalization gap. Even though this is important, it does not give a clear picture and the conclusions drawn from [1] might not be decisive.
>
> More precisely, they do not optimize the complexity measures directly (could be in the form of a regularizer), however, they train a standard network and observe the correlation. One of the reasons behind doing this is that it is hard to have a controlled environment where the impact of the complexity measure regularization and implicit regularization can be decoupled. However, in certain cases we do observe positive correlation between minimizing certain complexity measures and the generalization gap eg. L2 norm minimization via weight decay.
>
> These caveats are very clearly mentioned in [1] and is actually an interesting field of future research. A step towards that is our paper (we would also like to refer to [3]) where we directly optimize stable rank and observe positive correlation with generalization gap. Due to this difference, that we directly optimize the measure instead of just observing it, [1]’s conclusions do not apply here.
>
>
>
> Q2: Optimal value of stable rank
> Ans: Agreed, just like almost all regularization methods, there is no clear value that one can choose a priori for the desired stable rank.  We have thus tried to reduce the number of hyper-parameters to one hyper-parameter c.
>
> Q3: For the generalization of GANs, this paper [2] shows that the gradient of the optimal discriminator goes toward 0…
> Ans: Thank you very much for pointing this paper out. Even though we do not have a mathematical form linking gradient norm and low stable rank directly, empirically we did observe that low stable rank leads to discriminators with low gradient norm at the vicinity of real and generated samples (please refer Figs 15 and 16 in the Appendix). The empirical Lipschtiz also offers a very similar argument (Fig 5 in the main text, and Appendix E.2).
>
> [1] "Fantastic Generalization Measures and Where to Find Them"
> https://openreview.net/forum?id=SJgIPJBFvH&noteId=SJgIPJBFvH
> [2] "Improving Generalization and Stability of Generative Adversarial Networks"
> https://openreview.net/forum?id=ByxPYjC5KQ
>
> [3] “Spectral Norm Regularization for Improving the Generalizability of Deep Learning”, Yoshida and Miyato, 2017.

---

### Decision · Program_Chairs · 2019-12-19

**Decision:**

Accept (Spotlight)

**Comment:**

The authors propose stable rank normalization, which minimizes the stable rank of a linear operator and apply this to neural network training. The authors present techniques for performing the normalization efficiently and evaluate it empirically in a range of situations. The only issues raised by reviewers related to the empirical evaluation. The authors addressed these in their revisions.